# With Argus Eyes: Assessing Retrieval Gaps via Uncertainty Scoring to Detect and Remedy Retrieval Blind Spots

Zeinab Sadat Taghavi [1]   Ali Modarressi [2,3]   Hinrich Schütze [2,3]   Andreas Marfurt [1]

## Abstract

Reliable retrieval-augmented generation (RAG) systems depend fundamentally on the retriever's ability to find relevant information. We show that neural retrievers used in RAG systems have *blind spots*, which we define as the failure to retrieve entities that are relevant to the query, but have low similarity to the query embedding. We investigate the training-induced biases that cause such blind-spot entities to be mapped to inaccessible parts of the embedding space, resulting in low retrievability. Using a large-scale dataset constructed from Wikidata relations and first paragraphs of Wikipedia, and our proposed Retrieval Probability Score (RPS), we show that blind spot risk in standard retrievers (e.g., CONTRIEVER, REASONIR) can be predicted pre-index from entity embedding geometry, avoiding expensive retrieval evaluations. To address these blind spots, we introduce ARGUS, a pipeline that enables the retrievability of high-risk (low-RPS) entities through targeted document augmentation from a knowledge base (KB), first paragraphs of Wikipedia, in our case. Extensive experiments on BRIGHT, IMPLIRET, and RAR-B show that ARGUS achieves consistent improvements across all evaluated retrievers (averaging +3.4 nDCG@5 and +4.5 nDCG@10 absolute points), with substantially larger gains in challenging subsets. These results establish that preemptively remedying blind spots is critical for building robust and trustworthy RAG systems (Code and data: github.com/ZeinabTaghavi/With_Argus_Eyes)

**Note:** Part of this work was done while Z. Taghavi was at CIS, LMU Munich, and associated with MCML. [1]Lucerne University of Applied Sciences and Arts (HSLU) [2]Center for Information and Language Processing (CIS), Ludwig Maximilian University of Munich (LMU) [3]Munich Center for Machine Learning (MCML). Correspondence to: Zeinab Sadat Taghavi <zeinabsatad.taghavi@hslu.ch>.

*Proceedings of the 43rd International Conference on Machine Learning*, Seoul, South Korea. PMLR 306, 2026. Copyright 2026 by the author(s).

## 1. Introduction

Retrieval-augmented generation (RAG) has become a core building block of modern NLP systems, powering question answering, assistants, and tool-using agents by grounding generation in external evidence (Lewis et al., 2020; Guu et al., 2020; Gao et al., 2023). In these settings, trustworthiness hinges on a single component, which is the retriever's ability to surface the right information when it is needed (Gao et al., 2023). Otherwise, the retriever becomes a single point of failure for the overall pipeline, undermining robustness (Gao et al., 2023). Today's RAG pipelines increasingly rely on neural retrievers, which outperform lexical methods such as BM25 by capturing semantic similarity beyond surface word overlap (Robertson & Zaragoza, 2009; Karpukhin et al., 2020; Izacard et al., 2022). However, this shift toward semantic matching can introduce a new failure mode; retrieval depends on how queries and documents are positioned in the embedding space, and some relevant information can become systematically harder to retrieve due to unfavorable embedding geometry.

This geometric failure creates a silent bottleneck, compromising the retriever's robustness against specialized or semantically distant entities. Relevant evidence may exist in the corpus, but the retriever fails to surface it, causing the generator to fall back on ungrounded completions or hallucinations (Ji et al., 2023). We therefore ask whether such misses are merely random noise or instead reflect a deeper systematic phenomenon in neural retrieval. In particular, we study systematic blind spots: entity-centric gaps where certain entities (and the documents that mention them) are consistently missed even when they are relevant to the query, especially when relevance is semantic rather than driven by lexical overlap. Crucially, blind spots are relative to the retrieval budget; with a larger top-$k$ window, more evidence becomes accessible. But, relying on very large retrieval windows is often undesirable in RAG, because longer contexts can dilute the generator's effective focus and reduce downstream quality (e.g., "lost-in-the-middle" / "context rot") (Liu et al., 2024; Hong et al., 2025; Modarressi et al., 2025; Taghavi et al., 2025). Consequently, ensuring that relevant entities are geometrically accessible under practical budgets (e.g., $k \in [5, 50]$) is not only an efficiency consideration but

also fundamental for ensuring that RAG pipelines remain robust and trustworthy across diverse downstream tasks.

Existing retrieval evaluation and optimization are largely query-centric and post-hoc. They measure performance conditioned on a benchmark's queries, but do not reveal which entities are intrinsically hard to retrieve, nor do they support pre-deployment auditing of what a retriever will systematically miss (Bajaj et al., 2016; Thakur et al., 2021; Muennighoff et al., 2023). This limitation is amplified by the mismatch between domain-limited benchmarks and the broad, web-scale training of neural retrievers, where domain-specific test suites may overlook global failure patterns (Thakur et al., 2021; SU et al., 2025; Xiao et al., 2024). To obtain a complementary, domain-agnostic view, we construct a large random sample of Wikidata-Wikipedia aligned entities and use it to quantify entity-level retrievability risk (Vrandečić & Krötzsch, 2014; Wikimedia Foundation, 2025). Specifically, we introduce the Retrieval Probability Score (RPS), defined with respect to the user's top-$k$ budget. Formally, $\text{RPS}_k$ is the expected value of top-k hit probability over an entity's related query set that is derived from Wikidata Knowledge Graphs (KG) relations. This metric evaluates retrievability by measuring the frequency with which a target entity surfaces in the top-$k$ when ranked against a large pool of strictly disjoint neutral entities. These neutral candidates are randomly sampled Wikidata entities filtered to exclude all directly linked neighbors of both the target and query entities in the Wikidata KG, ensuring a controlled assessment of geometric robustness. Intuitively, $\text{RPS}_k$ represents retrieval consistency; for instance, an $\text{RPS}_k = 0.2$ indicates the entity is successfully retrieved for only 20% of its associated queries. Equivalently, a low RPS signals a high miss probability, a systematic blind spot, motivating the question of whether such failures can be predicted pre-index directly from embedding geometry.

Applying RPS at scale reveals substantial variation in retrievability. Across neural retrievers, entities range from consistently retrievable to persistently missed. Moreover, this variation is not arbitrary; when we assign entities to *low/mid/high* RPS terciles and visualize their embeddings using a two-dimensional projection, low- and high-RPS entities occupy geometrically distinguishable regions in embedding space, indicating structured blind-spot zones rather than random failures. As we increase the neutral pool size, these projections become more diagnostic. For robust retrievers, a larger fraction of entity embedding points remains in the high-RPS region, whereas for standard baselines, more entity points shift into and accumulate within low-RPS regions. Crucially, this geometric regularity implies that retrievability risk is encoded in the embeddings; we train lightweight diagnostic probes on entity embeddings labeled with empirical $\text{RPS}_k$, enabling pre-index prediction of high-risk entities without expensive retrieval simulations.

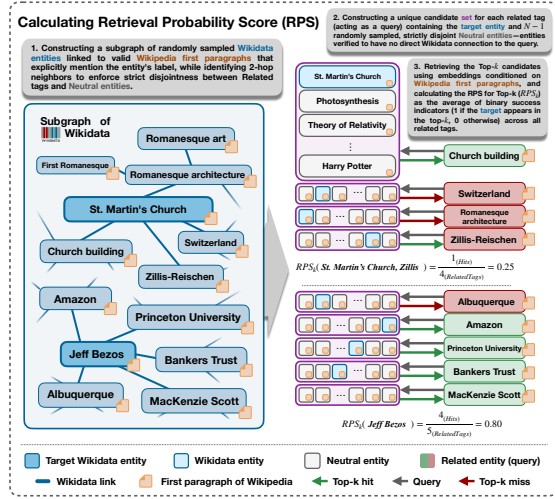

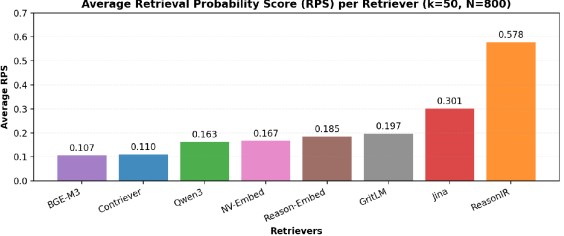

*Figure 1.* **Retrieval Probability Score (RPS) computation and retriever blind-spot analysis. (Top)** Evaluation pipeline: (1) construct a Wikidata–Wikipedia aligned dataset, (2) build query-specific retrieval sets with strictly disjoint neutral entities, and (3) compute $RPS$ from retrieval consistency. **(Bottom)** Average RPS over a large random entity sample at $k = 50$ with $N = 800$ neutrals (suppressing chance hits). Standard retrievers succeed only rarely (e.g., Contriever $\approx 0.11$), implying that for a random entity nearly 90% of valid top-$k$ retrieval opportunities fail.

Building on these findings, we propose ARGUS (**A**ssessing **R**etrieval **G**aps via **U**ncertainty **S**coring ), a diagnosis-to-remedy pipeline for retriever blind spots. ARGUS first predicts entity retrievability ($\text{RPS}_k$) under a target retriever and flags high-risk entities via thresholding ($\text{RPS}_k < \tau$). It then remedies these blind spots through targeted knowledge augmentation from a Reference KB, constructing augmented document views via either document expansion by concatenation or KB-guided LLM synthesis, and indexing these views alongside the original. Across BRIGHT, IMPLIRET, and RAR-B, ARGUS (Järvelin & Kekäläinen, 2002; SU et al., 2025; Xiao et al., 2024; Taghavi et al., 2025), yields consistent improvements in retrieval scores (nDCG@5/10) across eight popular neural retrievers.

**Contributions:** (i) We introduce RPS and a large-scale Wikidata-Wikipedia aligned protocol for assessing entity-level retrievability risk. (ii) We show that blind-spot risk is predictable from embedding representations, enabling pre-index detection via lightweight probes. (iii) We propose ARGUS, a practical remedy pipeline that augments high-

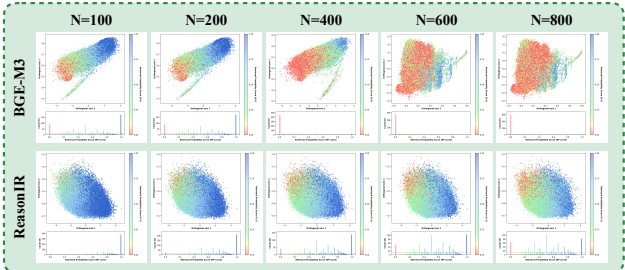

*Figure 2.* **LDA projections of entity embeddings labeled by RPS terciles (low/mid/high) at $k = 50$ under increasing neutral pool sizes $N$, comparing a low-RPS retriever (BGE-M3) to a high-RPS retriever (REASONIR-8B).** Robust retrievers retain denser high-RPS regions (blue) as $N$ grows, indicating higher expected top-$k$ retrievability for a random entity, while persistent low-RPS regions (red) across models confirm intrinsic blind spots.

risk entities through targeted KB context. (iv) ARGUS demonstrates robust nDCG gains across benchmarks and retriever architectures.

**Conflict of Interest Disclosure**  The authors declare that they have no financial conflicts of interest related to this work.

## 2. Related Work

**Neural Retrieval.** Neural information retrieval has largely shifted from lexical matching to dense retrieval, where a query encoder and a document encoder, that is often a dual-encoder architecture, map text into a shared embedding space and rank candidates by vector similarity (i.e., cosine similarity) (Karpukhin et al., 2020). This paradigm underlies many modern retriever families used in RAG pipelines, including general-purpose dense models and more specialized retrievers trained for stronger reasoning or supervision (e.g., BGE-M3, JINA-V3, and REASONIR-8B) (Chen et al., 2024; Sturua et al., 2024; Shao et al., 2025). Because retrieval decisions are mediated by the geometry of these learned representations, dense retrievers can succeed beyond surface overlap but also exhibit systematic behaviors tied to how entities and contexts are embedded (Izacard et al., 2022; Santhanam et al., 2022). Our work targets this setting, focusing on diagnosing and mitigating entity-level blind spots in neural retrievers.

**Reliable RAG and Pre-index Auditing.** In RAG settings, trustworthiness hinges on the retriever: when evidence is not surfaced, generators can hallucinate even if the knowledge exists in the corpus (Lewis et al., 2020; Shuster et al., 2021; Ji et al., 2023). While prior work strengthens the retrieval stack through query-side interventions or post-retrieval reranking (Nogueira & Cho, 2019; Karpukhin et al., 2020; Ma et al., 2023), these methods typically treat the index as given. Consequently, standard retrieval evalua-

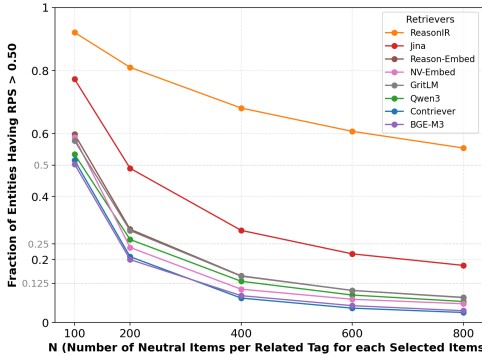

*Figure 3.* **Impact of neutral pool size ($N$) on fraction of entities with $\text{RPS}_k > 0.5$ ($k = 50$).** At $N = 100$, successful retrieval rates match the chance regime ($k/N \approx 0.5$). Beyond $N \geq 400$, curves decouple from chance and plateau, revealing stable, model-specific behavior. Hence, we adopt $N = 800$, so that high RPS reflects genuine geometric retrievability.

tion remains predominantly query-centric, relying on fixed benchmarks (e.g., MS MARCO, BEIR) to estimate average performance (Bajaj et al., 2016; Thakur et al., 2021). Although research has examined robustness to hard negatives, dataset bias, and distribution shifts, these approaches are largely post-hoc and require ground-truth queries to surface failures (Xiong et al., 2021; Yu et al., 2022; Mallen et al., 2023; Thakur et al., 2024). To enable pre-deployment auditing, we propose a shift to entity-centric risk estimation. By defining the RPS, we quantify intrinsic retrievability risk from embedding representations, allowing blind spots to be predicted and mitigated at indexing time.

## 3. Assessing Retriever Blind Spots

To distinguish intrinsic blind spots from idiosyncratic query effects, we audit neural retrievers at the *entity level*, asking how reliably an entity can be surfaced under a fixed top-$k$ budget across many query contexts. This motivates a large-scale, controlled measurement setup that decouples retrievability from query distributions.

### 3.1. Wikidata-Wikipedia Alignment for Retrievability Profiling

Leveraging the diverse relations in Wikidata, we construct a large-scale, domain-agnostic dataset for entity-centric retrievability auditing by aligning Wikidata's structured graph with Wikipedia's text through the following pipeline (Figure 1):

**(1) Entity Sampling.** We randomly sample $7 \times 10^6$ Wikidata entities and retain those with an English Wikipedia page, yielding a target set $X$. We apply lightweight cleaning and keep only entities with at least one valid related entity (details in Appendix B.1).

*Table 1.* **Predicting RPS from embedding geometry.** Reporting the best diagnostic probes that are selected based on the lowest RMSE, we observe high correlation (Pearson $r \approx 0.65\text{-}0.80$) and classification accuracy ($\approx 0.65\text{-}0.80$). This confirms that blind spots are geometrically encoded and detectable prior to indexing. (See Appendix C for full results of different model configurations.)

| Retriever | Architecture | Regression Metrics | | | | Semi-Classification Metrics | | | | | |
|---|---|---|---|---|---|---|---|---|---|---|---|
| | | RMSE ($\downarrow$) | MAE ($\downarrow$) | Pearson $r$ ($\uparrow$) | Spearman $\rho$ ($\uparrow$) | Macro-F1 ($\uparrow$) | Macro-Rec. ($\uparrow$) | Macro-Prec. ($\uparrow$) | Prec$_\text{weighted}$ ($\uparrow$) | F1$_\text{weighted}$ ($\uparrow$) | Accuracy ($\uparrow$) |
| BGE-M3 | XGBoost | 0.168 | 0.118 | 0.681 | 0.644 | 0.658 | 0.540 | 0.573 | 0.781 | 0.762 | 0.781 |
| CONTRIEVER | XGBoost | 0.157 | 0.109 | 0.658 | 0.622 | 0.727 | 0.501 | 0.506 | 0.795 | 0.777 | 0.795 |
| QWEN3-EMBEDDING | XGBoost | 0.153 | 0.111 | 0.781 | 0.721 | 0.699 | 0.619 | 0.646 | 0.764 | 0.760 | 0.764 |
| NV-EMBED | XGBoost | 0.173 | 0.124 | 0.640 | 0.595 | 0.657 | 0.517 | 0.541 | 0.757 | 0.740 | 0.757 |
| REASON-EMBED | XGBoost | 0.156 | 0.114 | 0.764 | 0.742 | 0.688 | 0.595 | 0.609 | 0.752 | 0.748 | 0.752 |
| GRITLM-7B | XGBoost | 0.157 | 0.115 | 0.745 | 0.677 | 0.682 | 0.595 | 0.620 | 0.762 | 0.754 | 0.762 |
| JINA-V3 | Ridge | 0.178 | 0.137 | 0.667 | 0.659 | 0.641 | 0.557 | 0.574 | 0.655 | 0.653 | 0.655 |
| REASONIR-8B | Ridge | 0.156 | 0.121 | 0.779 | 0.788 | 0.710 | 0.658 | 0.674 | 0.674 | 0.674 | 0.674 |

**(2) Context Grounding.** For each $x \in X$, we use the first paragraph of its Wikipedia page as the canonical context $w_x$, and enforce that the entity's Wikidata surface form appears in $w_x$, otherwise we minimally prepend it as a separate span at the beginning of the text, so mention-based pooling is well-defined, and alignments remain clean (Appendix B.1).

**(3) Related Entities (Query Construction).** From Wikidata 1-hop relations, we derive a set of related entities $\mathcal{T}_x$ that have English Wikipedia pages. We treat each related entity's Wikidata surface form as the query, but compute a context-conditioned query embedding by encoding its Wikipedia first paragraph to reduce ambiguity from polysemous labels (e.g., there are many "St. Martin's" churches). These related entities serve as proxy queries capturing contexts in which $x$ should be retrievable (e.g., *St. Martin's Church → Romanesque architecture*).

**(4) Neutral Baseline (Controlled Competition).** For each related entity $t \in \mathcal{T}_x$, we construct a related-entity-specific neutral pool $\mathcal{Z}_\text{neut}(t)$ of size $N$, where each $z \in \mathcal{Z}_\text{neut}(t)$ is a neutral entity, and enforce KG disjointness, i.e., each neutral $z$ is not directly linked to $t$ in Wikidata (no 1-hop edge under any property). As with targets, each neutral item is represented by its Wikipedia first paragraph, and we require its Wikidata surface form to be explicitly mentioned in that paragraph. This yields a controlled setting in which failures are less attributable to semantic ambiguity and more indicative of the retriever's embedding geometry. We next define RPS over these related-entity-specific pools, treating $k$ as the user-defined retrieval budget, and later select a conservative $N$ to ensure RPS is stable and not driven by random hits.

### 3.2. Retrieval Probability Score ($RPS$)

We quantify entity-level retrievability under a fixed top-$k$ budget using the *Retrieval Probability Score* (RPS). Let $E_\theta(\cdot)$ denote the target retriever encoder (token-level when available), and let $g(\cdot)$ denote the retriever-specific pooling operator that extracts a single mention-aware vector from these representations given a mention span. We represent an entity $u$ by an embedding $\mathbf{e}_u = g(E_\theta(w_u), s_u) \in \mathbb{R}^h$, where $w_u$ is the Wikipedia first paragraph of $u$ and $s_u$ is the

span of $u$'s Wikidata surface-form mention within $w_u$. We always encode the full paragraph $w_u$; the span $s_u$ is used only by $g(\cdot)$ to extract a mention-aware pooled embedding (details in Appendix B.4).

We apply the same encoding procedure to all entities. Each related entity $t_i \in \mathcal{T}_x$ yields a query embedding $\mathbf{q}_i = \mathbf{e}_{t_i}$ from its Wikipedia first paragraph with pooling at its mention span, and the target $x$ and neutrals $z_{i,j} \in \mathcal{Z}_\text{neut}(t_i)$ yield candidate embeddings $\mathbf{e}_x$ and $\mathbf{e}_{z_{i,j}}$ computed identically from their own paragraphs and mention spans.

**Controlled retrieval pools.** For each related entity $t_i \in \mathcal{T}_x$, we form a candidate set of size $N$ by placing the target entity $x$ alongside $N-1$ neutrals:

$$\mathcal{C}_i = \{x\} \cup \{z_{i,1}, \ldots, z_{i,N-1}\}, \quad z_{i,j} \sim \text{Uniform}(\mathcal{Z}_\text{neut}(t_i)).$$

We rank candidates $c \in \mathcal{C}_i$ by cosine similarity $\cos(\mathbf{q}_i, \mathbf{e}_c)$ and define a top-$k$ *hit* as:

$$\text{Hit}_\text{k}(x, t_i) = \mathbb{I}\big[\text{rank}(x \mid t_i, \mathcal{C}_i) \leq k\big].$$

**RPS definition and interpretation.** We define RPS of entity $x$ as the expected value of a successful hit across the distribution of all potentially relevant facts or queries $\mathcal{P}(\mathcal{T}_x)$ associated with it:

$$\text{RPS}_k(x \mid w_x) = \mathbb{E}_{t \sim \mathcal{P}(\mathcal{T}_x)}\big[\text{Hit}_\text{k}(x, t)\big].$$

In practice, we approximate this expectation via the empirical average hit rate across the sampled related entities:

$$\text{RPS}_k(x \mid w_x) \approx \frac{1}{|\mathcal{T}_x|} \sum_{t_i \in \mathcal{T}_x} \text{Hit}_\text{k}(x, t_i).$$

Intuitively, $\text{RPS}_k(x \mid w_x)$ represents the probability of discovery. Low RPS implies a high expected miss probability, which is a blind spot, under the given top-$k$ budget, whereas high RPS indicates consistent geometric discoverability in the retriever's embedding space.

**Geometric Structure of Blind Spots.** Applying RPS at scale reveals substantial variation in retrievability: standard retrievers such as CONTRIEVER exhibit consistently low average scores, whereas robust models like ReasonIR surface

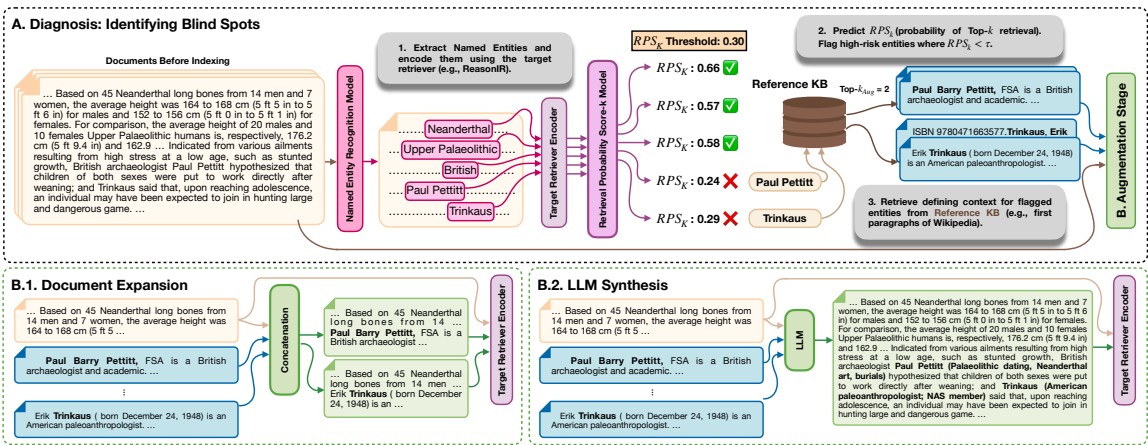

*Figure 4.* **The ARGUS Pipeline: Diagnosis and Remedy of Geometric Blind Spots.** (A) Diagnosis: The system first extracts named entities and predicts their retrievability ($RPS_k$) using the target retriever. Entities falling below the safety threshold ($RPS < \tau$) are flagged as blind spots (high-risk) located in inaccessible regions of the embedding space. (B) Augmentation: To remedy these blind spots, ARGUS retrieves defining context from a Reference KB. We employ two strategies, (B.1) Document Expansion (Concatenation) or (B.2) LLM Synthesis, to generate augmented document views. By indexing these views alongside the original, we enable the retrievability of previously unknown entities.

significantly more entities (Figure 1, bottom). To confirm these failures are structural rather than random, we visualize entity embeddings via LDA after partitioning entities into RPS terciles. The projections reveal a clear geometric spectrum where low-RPS entities (red) cluster into distinct blind spot regions separable from high-RPS areas (blue), a pattern that can be seen in their histogram and persists even in robust architectures (Figure 2). The complete set of projections for all retrievers is provided in Appendix B.5.

**Neutral Pool Sufficiency.** Notably, as neutral competition increases, the low-RPS mass becomes more pronounced for standard retrievers, raising the question of which pool size $N$ yields a stable audit. We analyze the fraction of entities with $RPS_k > 0.5$ under increasing $N$ at our maximum budget $k = 50$ (Figure 3). The setting of $k = 50$ is the most challenging for controlling randomness, as the chance baseline scales with $k/N$. At small $N$, the chance hit rate is high ($k/N$ large), so observed success can be inflated by random collisions and is not diagnostic of true retrievability; however, beyond $N \approx 400$, the curves decouple and plateau, indicating stability. Accordingly, we adopt $N = 800$ as a conservative setting to ensure that measured blind spots reflect predictable geometric failures suitable for diagnosis and remedy.

# 4. Detecting Blind Spots from Embedding Geometry

Since blind spots occupy distinct geometric regions (Section 3), we hypothesize that retrievability risk is an intrinsic property that can be estimated directly from an entity's representation, without expensive retrieval simulations.

## 4.1. Diagnostic Probes for RPS Prediction

We formulate blind-spot detection as a supervised regression task, learning a diagnostic function that maps an entity embedding $\mathbf{e}_x$ to a predicted retrievability score $\widehat{\text{RPS}}_k(x \mid w_x) \in [0, 1]$. Since $x$ is represented in its canonical Wikipedia context $w_x$ (Section 3.2), this task corresponds to predicting the retrievability score directly from the embedding $\mathbf{e}_x$. Because embedding geometries vary across architectures, we train a separate probe for each retriever, exploring three model families: linear Ridge Regression, non-linear tree-based models (XGBoost), and Multi-Layer Perceptrons (MLP).

**Training and Selection.** Probes are trained on entity embeddings labeled with empirical $\text{RPS}_k(x \mid w_x)$ computed with $N{=}800$ neutrals for a fixed retrieval budget $k$. We employ a standard train/validation/test split and select hyperparameters by minimizing RMSE on the validation set. We evaluate a comprehensive sweep of probe configurations (detailed in Appendix C.1); for clarity, we report only the best-performing probe for each retriever in Table 1. The resulting predictor enables a pre-index audit; by thresholding $\widehat{\text{RPS}}_k(x \mid w_x) < \tau$, we can flag high-risk entities solely from their vector representations. While $\tau$ can be set based on application sensitivity, throughout this paper we use a fixed global threshold $\tau{=}0.3$ (slightly below the lowest-tercile cutoff, $\approx 0.33$) across all experiments.

## 4.2. Regression and Semi-Classification Performance

Table 1 reports the best-performing probe for each neural retriever, demonstrating strong agreement with empirical $\text{RPS}_k(x \mid w_x)$ (e.g., Pearson $r \approx 0.65\text{-}0.80$) and low pre-

*Table 2.* **Downstream retrieval performance (nDCG@5/10) on BRIGHT, IMPLIRET, and RAR-B.** We compare standard baselines against ARGUS remedies, Document Expansion, and LLM Synthesis, across eight neural retrievers. The Full Benchmark Avg. columns report mean performance over the complete task suite for each benchmark (e.g., 10 of the BRIGHT domains), rather than only the representative subsets shown. ARGUS yields robust gains across retrievers, supporting the efficacy of remedying geometric blind spots by targeted augmentations. (See Appendix F for the full per-task breakdown.) **Colors**: **Best Result**, 2nd Best, 3rd Best.

| | | Bright | | | | Impliret | | | Rar-b | | | |
| | | *Shown (3/10)* | | | *Full Benchmark Avg.* | *Shown (2/2)* | | *Full Benchmark Avg.* | *Shown (3/7)* | | | *Full Benchmark Avg.* |
| Retriever | Augmentation | Biology | Econ | Sust | *Avg. (All 10 Tasks)* | Multi | Uni | *Avg. (All 2 Tasks)* | ARC | PIQA | Hella | *Avg. (All 7 Tasks)* |
|---|---|---|---|---|---|---|---|---|---|---|---|---|
| BGE-M3 | Baseline | 7.8/9.5 | 10.0/11.7 | 9.0/10.1 | 10.2/11.0 | 17.9/23.3 | 13.4/18.8 | 15.7/21.1 | 7.8/9.0 | 20.9/22.9 | 23.3/25.5 | 17.1/19.5 |
| | ARGUS (Doc-Exp.) | 8.6/11.4 | **13.1/17.6** | 10.4/12.8 | 12.5/**15.9** | 30.2/38.3 | 26.7/34.0 | 28.4/36.1 | 7.8/9.2 | 21.9/24.7 | 24.0/26.9 | 17.4/20.0 |
| | ARGUS (LLM-Synth.) | 13.6/14.7 | 12.6/13.9 | 12.1/13.1 | 14.3/15.3 | 22.5/27.5 | 16.5/21.5 | 19.5/24.5 | 8.5/9.8 | 21.8/23.8 | 24.8/26.8 | 18.2/20.0 |
| CONTRIEVER | Baseline | 7.2/9.2 | 10.7/10.5 | 7.1/8.9 | 9.0/9.8 | 12.8/18.3 | 10.4/15.2 | 11.6/16.8 | 7.4/8.6 | 23.1/25.1 | 24.1/26.4 | 21.9/23.7 |
| | ARGUS (Doc-Exp.) | 8.6/13.0 | 13.7/16.4 | 8.1/11.9 | 11.6/14.9 | 18.0/24.9 | 16.9/22.8 | 17.4/23.9 | 7.4/9.0 | 24.1/26.8 | 24.3/26.8 | 24.4/27.9 |
| | ARGUS (LLM-Synth.) | 8.8/11.5 | 10.1/11.2 | 7.1/8.2 | 10.2/11.8 | 18.4/24.3 | 17.0/22.4 | 17.7/23.3 | 7.7/9.7 | 27.2/30.6 | 28.5/32.8 | 25.6/29.2 |
| QWEN3-EMBEDDING | Baseline | 10.5/12.4 | 11.7/12.7 | 9.2/10.2 | 10.0/10.8 | 8.0/10.8 | 3.9/5.3 | 5.9/8.0 | 7.4/8.8 | 17.1/19.4 | 21.8/24.0 | 14.5/16.2 |
| | ARGUS (Doc-Exp.) | 18.5/27.1 | 15.2/16.2 | 9.8/13.4 | 13.2/16.6 | 10.4/15.1 | 5.5/7.3 | 8.0/11.2 | 7.9/9.3 | 18.7/22.1 | 26.5/31.9 | 15.8/18.5 |
| | ARGUS (LLM-Synth.) | 9.1/13.7 | 14.2/15.2 | 10.7/11.7 | 12.1/13.7 | 8.4/12.0 | 5.0/5.4 | 6.7/8.7 | 7.9/9.5 | 19.5/22.3 | 25.4/28.7 | 16.5/18.4 |
| NV-EMBED-V2 | Baseline | 14.0/16.5 | 12.2/13.2 | 9.7/10.7 | 11.9/13.1 | 33.9/38.5 | 24.3/29.2 | 29.1/33.9 | 14.3/16.2 | 34.8/37.6 | 33.7/36.2 | 21.4/24.3 |
| | ARGUS (Doc-Exp.) | 17.7/19.2 | 15.2/16.7 | 11.7/12.7 | 15.5/16.7 | 50.2/53.2 | 40.2/43.2 | 45.2/48.2 | 16.2/17.2 | 38.2/39.2 | 36.2/37.2 | 24.2/25.5 |
| | ARGUS (LLM-Synth.) | 15.7/17.7 | 13.2/14.7 | 10.2/11.7 | 13.6/15.3 | 38.2/42.2 | 28.2/32.2 | 33.2/37.2 | 15.2/17.5 | 36.0/39.0 | 35.0/37.5 | 22.6/25.6 |
| REASON-EMBED | Baseline | 14.5/18.6 | 10.2/11.2 | 10.5/12.4 | 11.1/12.6 | 7.8/11.0 | 1.8/2.4 | 4.8/6.7 | 7.9/9.2 | 12.4/14.1 | 20.8/22.9 | 12.9/14.3 |
| | ARGUS (Doc-Exp.) | 15.9/22.4 | 10.4/11.5 | 11.2/13.4 | 13.8/16.4 | 8.2/12.2 | 2.0/2.8 | 5.1/7.5 | 8.1/9.6 | 13.0/15.0 | 23.2/26.7 | 14.3/16.1 |
| | ARGUS (LLM-Synth.) | 17.5/22.8 | 12.0/13.7 | 12.7/14.0 | 13.5/15.4 | 8.0/11.5 | 3.0/3.4 | 5.5/7.4 | 8.2/10.1 | 13.5/15.6 | 24.8/28.2 | 14.4/16.2 |
| GRITLM-7B | Baseline | 5.9/7.0 | 4.1/4.4 | 4.1/4.8 | 5.7/6.4 | 5.6/7.3 | 15.2/16.7 | 10.4/12.0 | 3.1/3.9 | 3.3/3.9 | 16.5/18.3 | 10.3/11.5 |
| | ARGUS (Doc-Exp.) | 6.7/9.3 | 4.8/5.0 | 5.2/6.9 | 7.1/8.4 | 6.1/8.6 | 22.2/24.2 | 14.2/16.4 | 3.2/4.0 | 3.4/4.0 | 19.0/22.1 | 11.4/12.8 |
| | ARGUS (LLM-Synth.) | 9.5/10.2 | 7.2/7.8 | 6.2/7.0 | 8.2/9.0 | 5.9/8.0 | 16.5/18.0 | 11.2/13.0 | 3.6/4.2 | 3.9/4.5 | 18.5/20.5 | 11.4/13.0 |
| JINA-V3 | Baseline | 12.0/15.2 | 18.6/19.2 | 13.1/15.8 | 14.2/16.1 | 14.8/20.6 | 10.9/15.2 | 12.9/17.9 | 11.1/13.2 | 26.5/29.1 | 24.6/27.0 | 15.9/17.8 |
| | ARGUS (Doc-Exp.) | 12.8/16.5 | 19.9/21.6 | 14.3/18.2 | 15.7/18.5 | 17.5/24.2 | 12.2/17.3 | 14.8/20.8 | 11.3/13.6 | 27.5/30.9 | 27.4/31.4 | 16.7/19.2 |
| | ARGUS (LLM-Synth.) | 13.4/16.2 | 20.2/21.2 | 14.7/17.0 | 15.7/17.4 | 16.2/19.7 | 10.7/13.2 | 13.4/16.4 | 12.0/14.2 | 27.5/30.0 | 26.0/28.5 | 16.9/18.9 |
| REASONIR-8B | Baseline | 16.6/19.1 | 13.9/16.5 | 10.2/11.4 | 13.6/15.0 | 22.7/27.7 | 8.1/10.5 | 15.4/19.1 | 12.2/13.5 | 23.3/25.9 | 30.9/33.2 | 18.6/20.1 |
| | ARGUS (Doc-Exp.) | 18.0/24.4 | 16.6/22.2 | 12.2/13.8 | 17.3/21.4 | 30.8/42.0 | 12.5/19.2 | 21.7/30.6 | 12.2/13.5 | 26.2/27.3 | 34.3/35.8 | 20.8/21.9 |
| | ARGUS (LLM-Synth.) | 20.0/25.2 | 12.7/16.8 | 11.3/12.8 | 15.8/18.3 | 25.8/32.0 | 8.7/11.3 | 17.3/21.7 | 12.9/14.5 | 25.7/26.8 | 33.8/35.3 | 20.4/21.6 |

diction error under the regression objective. To translate these scores into actionable risk categories, we also evaluate a *semi-classification* view by discretizing entities into three $\text{RPS}_k(x \mid w_x)$ bands: *low* ([0, 0.33)), *mid* ([0.33, 0.66)), and *high* ([0.66, 1]), and measuring how well probes recover these categories. Across retrievers, probes achieve strong performance on this task (accuracy $\approx$ 0.65-0.80 with correspondingly high macro-F1), indicating that they reliably separate low-$\text{RPS}_k(x \mid w_x)$ entities from partially and highly retrievable ones. We further assess calibration to ensure predictions are not systematically biased; Appendix C.2 (Figure 7) shows predicted-empirical $\text{RPS}_k(x \mid w_x)$ densities and residuals that are well-centered around zero across retrievers, with density skew largely reflecting the natural imbalance of $\text{RPS}_k(x \mid w_x)$ in standard models. Overall, these results establish that blind-spot risk is detectable pre-index, enabling threshold-based flagging as the first stage of ARGUS.

## 5. ARGUS: Remedying Retriever Blind Spots

Building on Section 4, where we showed that RPS is predictable from embedding geometry via lightweight probes, we now introduce ARGUS, an offline pre-index time intervention for remedying retriever blind spots in IR/RAG corpora, requiring neither query rewriting nor expensive retrieval evaluation over the target corpus. ARGUS proceeds in two stages: **Diagnosis** flags high-risk named entities in each document using predicted $\widehat{\text{RPS}}_k$, and **Remedy** injects external defining context to construct augmented document

views for flagged entities and enhance their retrievability (Figure 4)

### 5.1. Diagnosis: Pre-Index Risk Estimation

Given a corpus $\mathcal{D}$ of documents, our goal is to identify high-risk *named-entities* that are likely to be blind spots for a target neural retriever under a fixed top-$k$ budget. We use named entities to denote an NER-extracted surface-form span in a document $d \in \mathcal{D}$. Concretely, we first run an off-the-shelf NER tagger to extract entity spans $s$ and their corresponding strings $m$ (details in Appendix D.1). For each extracted entity mention $m$ with span $s$ in document $d$, we compute a context-dependent embedding using the target retriever encoder and the same span pooling operator as in Section 3.2: $\mathbf{e}_{m,d} = g(E_\theta(d), s)$. We then apply the retriever-specific diagnostic model (Section 4.1), to estimate contextual retrievability of $m$: $\widehat{\text{RPS}}_k(m \mid d)$.

If an entity appears multiple times in $d$, we score each occurrence and assign the entity the minimum predicted score across named entities (risk-conservative); we then augment each flagged entity. We label $m$ as high-risk when $\widehat{\text{RPS}}_k(m \mid d) < \tau$ (default $\tau = 0.3$), and output the set of flagged entities

$$\mathcal{E}_{\text{risk}}(d) = \{\, m \,:\, \widehat{\text{RPS}}_k(m \mid d) < \tau \,\}.$$

This diagnosis stage is fully offline and lightweight; it requires only NER, document encoding, and probe inference (Figure 4, Diagnosis), and now, we can go over the remedy.

*Table 3.* Average extracted versus augmented entities per document. Percentages indicate the ratio of augmented to extracted entities. ARGUS augmentation remains sparse across retrievers and benchmarks, limiting index growth.

| Dataset | BRIGHT (10/10) | ImpliRet (2/2) | RAR-b (7/7) |
|---|---|---|---|
| Extracted | 1.01 | 3.12 | 0.43 |
| bge-m3 | 0.96 (95%) | 2.41 (77%) | 0.41 (96%) |
| gritlm | 0.83 (82%) | 2.48 (80%) | 0.36 (83%) |
| qwen3 | 0.86 (85%) | 1.51 (49%) | 0.37 (87%) |
| reasonir | 0.58 (57%) | 1.61 (52%) | 0.25 (58%) |

## 5.2. Remedy: Targeted Knowledge Augmentation

**Reference KB retrieval.** Note that while the Wikipedia first paragraphs in Section 3 were used for auditing (via label-grounded Wikidata alignment), here they serve as a retrievable reference KB for augmentation. Given the diagnosed set of high-risk entities $\mathcal{E}_{\text{risk}}(D)$, ARGUS injects a concise defining context at indexing time. For each flagged entity mention $m \in \mathcal{E}_{\text{risk}}(D)$, we query the Reference KB (Wikipedia first paragraphs) using the surface form of $m$ and retrieve the top $k_{\text{Aug}}$ passages with a fast lexical retriever (BM25S) (Lù, 2024). We set $k_{\text{Aug}}{=}2$, which provides sufficient disambiguating context to anchor the entity while keeping the augmentation lightweight.

We instantiate two augmentation strategies that trade off index growth against computation.

**1. Document Expansion by Concatenation** creates one augmented view per retrieved KB passage $p$ by appending it to the original document: $d_{m,p}^{\text{exp}} = d \parallel p$. Let $N_d = |\mathcal{E}_{\text{risk}}(d)|$ denote the number of flagged entities in $d$. This yields $1 + k_{\text{Aug}} \cdot N_d$ indexed views (the original $d$ plus one expansion per $(m, p)$ pair).

**2. KB-guided LLM Synthesis** instead aggregates all retrieved KB passages for all flagged entities in $d$ and prompts an LLM with $(d, \{p\})$ to produce a single unified augmented document $d^{\text{synth}}$ that inserts short, entity-focused clarifications only where necessary after the entity surface $m$ in the parentheses (prompting details in Appendix D.3). This option indexes exactly two views per document: the original $d$ and $d^{\text{synth}}$, reducing index growth at the cost of LLM computation.

In both cases, we index augmented views next to the original documents; we never replace them, hence, the corpus semantics are preserved while adding retrievable "support views" for previously high-risk entities (Figure 4, *B*).

ARGUS converts pre-index blind spot risk estimates into targeted pre-indexing time augmentations that improve downstream retrieval by remedying entity-level blind spots. In the next section, we evaluate this pipeline across multiple retrievers and benchmarks to quantify its impact under diverse retrieval settings.

*Table 4.* Performance under constrained augmentation budgets. The budget limits the number of additional indexed augmented views relative to the original corpus size. Entities are prioritized by predicted blind-spot risk (lowest RPS first).

| Model | Budget(%) | BRIGHT (3/10) | ImpliRet (2/2) | RAR-b (3/7) |
|---|---|---|---|---|
| bge-m3 | Baseline | 10.43 | 21.05 | 19.13 |
| | 20% | 10.81(+3.6%) | 21.82(+3.7%) | 21.73(+13.6%) |
| | 40% | 11.60(+11.2%) | 23.63(+12.3%) | 24.94(+30.4%) |
| | 60% | 12.65(+21.3%) | 26.10(+24.0%) | 25.53(+33.4%) |
| | 80% | 13.23(+26.8%) | 26.64(+26.6%) | 25.53 |
| | Full | 13.93(+33.6%) | 36.15(+71.8%) | 25.53 |
| ReasonIR | Baseline | 15.66 | 19.10 | 24.10 |
| | 20% | 16.57(+5.8%) | 19.69(+3.1%) | 25.13(+4.3%) |
| | 40% | 18.92(+20.8%) | 21.02(+10.1%) | 25.53(+5.9%) |
| | 60% | 21.13(+34.9%) | 22.95(+20.2%) | 25.53 |
| | 80% | 21.13 | 23.36(+22.3%) | 25.53 |
| | Full | 21.13 | 30.60(+60.2%) | 25.53 |

## 6. Experimental Setup

We evaluate whether ARGUS turns pre-index risk detection into downstream retrieval gains across multiple benchmarks and dense neural retrievers.

**Evaluation Benchmarks. Benchmarks:** We evaluate on BRIGHT (SU et al., 2025), IMPLIRET (Taghavi et al., 2025), and RAR-B (Xiao et al., 2024), covering multiple tasks/settings per benchmark (BRIGHT: 10 domains, IMPLIRET: 2 settings of "Word Knowledge" category, RAR-B: 7 subsets). **Metrics:** We report nDCG@5 and nDCG@10 in Table 2; additional cutoffs (e.g., nDCG@20/50) are provided in Appendix F.

**Retrievers and Baselines.** We test eight retrievers: BGE-M3, CONTRIEVER, QWEN3-EMBEDDING, NV-EMBED-V2, REASON-EMBED, GRITLM-7B, JINA-V3, REASONIR-8B (Chen et al., 2024; Izacard et al., 2022; Zhang et al., 2025; Lee et al., 2025; Chen et al., 2025; Muennighoff et al., 2025; Sturua et al., 2024; Shao et al., 2025). We compare Original indexing against ARGUS with *Document Expansion* or *LLM Synthesis*. We enforce a strict retriever-consistent pipeline; when evaluating a model, that retriever is used for embedding, risk diagnosis, and final ranking. The only exception is the internal lookup over the Reference KB, which we perform with BM25S as a lightweight, fast baseline (Lù, 2024).

**Implementation Details. ARGUS:** We extract candidate named entities using `dslim/bert-base-NER` and set the risk threshold to $\tau = 0.3$ on $\widehat{\text{RPS}}_k$. For each flagged entity $m$, we retrieve the top $k_{\text{Aug}} = 2$ passages from a Reference KB consisting of Wikipedia first paragraphs using BM25S (Lù, 2024). We query using the entity surface form and use the retrieved passages to construct augmented document views. We index these views alongside the original documents and run retrieval over the

*Table 5.* LLM augmentation without increasing index size. Instead of indexing additional augmented views, we replace each original document with its LLM-augmented version. ARGUS still improves retrieval quality, indicating that the gains are not solely due to index growth.

| Model | Method | BRIGHT (3/10) | ImpliRet (2/2) | RAR-b (3/7) |
|---|---|---|---|---|
| BGE-M3 | Baseline | 10.43 | 21.05 | 19.13 |
| | ARGUS (LLM Replacement) | 13.48 | 23.72 | 20.06 |
| | ARGUS (LLM Synthesis) | **13.90** | **24.50** | **20.13** |
| ReasonIR | Baseline | 15.66 | 19.10 | 24.10 |
| | ARGUS (LLM Replacement) | 17.86 | 21.15 | 25.49 |
| | ARGUS (LLM Synthesis) | **18.26** | **21.70** | **25.53** |

*Table 6.* Sensitivity of ARGUS to the blind-spot threshold $\tau$. $\tau = 1.0$ corresponds to augmenting all extracted entities, while $\tau = 0.0$ corresponds to the baseline without augmentation. Performance remains relatively stable around $\tau \in [0.3, 0.4]$.

| Model | Threshold ($\tau$) | BRIGHT (3/10) | ImpliRet (2/2) | RAR-b (3/7) |
|---|---|---|---|---|
| bge-m3 | Baseline (0.0) | 10.43 | 21.05 | 19.13 |
| | 0.2 | 12.52 | 30.01 | 19.81 |
| | ARGUS (0.3) | **13.93** | **36.15** | **20.26** |
| | 0.4 | 13.87 | 35.92 | 20.24 |
| | 0.6 | 13.91 | 35.05 | 20.25 |
| | 0.8 | 13.65 | 34.04 | 20.21 |
| | All (Entity) (1.0) | 13.31 | 33.45 | 20.19 |
| ReasonIR | Baseline (0.0) | 15.66 | 19.10 | 24.10 |
| | 0.2 | 18.91 | 25.96 | 24.94 |
| | ARGUS (0.3) | **21.13** | **30.60** | **25.53** |
| | 0.4 | 20.98 | 30.43 | 25.51 |
| | 0.6 | 20.59 | 30.21 | 25.42 |
| | 0.8 | 20.23 | 29.42 | 25.37 |
| | All (Entity) (1.0) | 19.87 | 28.76 | 25.21 |

expanded index using the same target retriever and scoring/ranking procedure as Original. **LLM synthesis:** We use QWEN3 (Qwen/Qwen3-30B-Instruct-2507 (Team, 2025)) to generate one synthesized view per document (Appendix D.3). **Probe:** For each retriever (and retrieval budget $k$), we use the best-performing probe from Section 4, selected by validation RMSE.

# 7. Experimental Results

We evaluate ARGUS across eight neural retrievers on three benchmarks. For readability, several tables report representative subsets of each benchmark rather than the full task suite. Specifically, we report 3/10 BRIGHT domains, both ImpliRet settings, and 3/7 RAR-b subsets and the full benchmark averages in Table 2. The complete per-task results are provided in Table 10 of Appendix F. Overall, targeted pre-index augmentation yields broad improvements in retrieval quality (nDCG@5/10) across diverse architectures and benchmarks.

## 7.1. End-to-End Retrieval Improvements

We evaluate ARGUS, via *Document Expansion* and *LLM Synthesis* (Figure 4), against the Original baseline across eight neural retrievers on BRIGHT IMPLIRET, and RAR-B. As we see in Table 2, averaged over nDCG@5 and nDCG@10 across all retrievers, Document Expansion improves the Full Benchmark Avg. by **+3.44** on BRIGHT, **+6.76** on IMPLIRET, and **+1.68** on RAR-B, while LLM Synthesis yields gains of **+2.44**, **+2.21**, and **+1.81**, respectively. Full per-task results appear in Table 10 of Appendix F. As an auxiliary diagnostic, Table 9 of Appendix E.1 compares the maximum entity RPS in retrieved vs. unretrieved gold documents, suggesting an association between entity-level retrievability and retrieval outcomes in some subsets. These results show that targeted index-time augmentation guided by the blind-spot predictors validated in Table 1 translates into tangible end-to-end retrieval gains under practical top-$k$ settings.

## 7.2. Document Expansion vs. LLM Synthesis

The two ARGUS strategies expose a practical trade-off between retrieval stability and index efficiency, while sharing the core benefit of preserving the original corpus (augmented views are indexed alongside original document $D$).

**Stability vs. Efficiency.** Document Expansion is the most consistent intervention in our experiments by appending retrieved KB contexts, it improves the Full Benchmark Avg. in most configurations, with particularly strong suite-level gains on IMPLIRET (**+6.76** averaged over nDCG@5/10) and solid improvements on BRIGHT (**+3.44**) and RAR-B (**+1.68**). In contrast, LLM Synthesis adds only one additional view per document (rather than growing with the number of flagged entities) while achieving competitive suite-level improvements, especially on BRIGHT (**+2.44**) and RAR-B (**+1.81**). Interestingly, on RAR-B, synthesis can outperform expansion for some standard retrievers such as CONTRIEVER (nDCG@10: 29.2 vs. 27.9), suggesting that coherent synthesized views may sometimes align better with query semantics than raw concatenation.

**Retriever Sensitivity.** Synthesis also exhibits higher variance, consistent with some retrievers being more sensitive to the structure and style of generated augmentations. For example, on IMPLIRET, JINA-V3 degrades with synthesis (nDCG@10: 17.9 → 16.4) but improves with expansion (20.8). Since both strategies augment the same flagged entities using the same retrieved KB evidence, this divergence suggests that augmentation *form* can interact with retriever behavior. Overall, ARGUS supports flexible deployment: LLM Synthesis for constrained index budgets, or Document Expansion for maximum stability and overall performance.

## 7.3. Efficiency and Budget-Constrained Augmentation

We next analyze the indexing overhead introduced by targeted augmentation. Table 3 shows that under the default threshold ($\tau = 0.3$), augmentation remains sparse, often

*Table 7.* Ablation of entity-aware and targeted augmentation strategies. ARGUS consistently outperforms untargeted and random alternatives, suggesting that both entity grounding and low-RPS selection are important for improving retrievability.

| Retriever | Method | BRIGHT (3/10) | ImpliRet (2/2) | RAR-b (3/7) |
|---|---|---|---|---|
| | Baseline | 10.43 | 21.05 | 19.13 |
| | Untargeted Random KB | 10.92 | 21.74 | 20.01 |
| BGE-M3 | Random Entity Augmentation | 11.48 | 23.10 | 21.22 |
| | All Entities | 12.63 | 31.42 | 24.40 |
| | ARGUS (Low-RPS) | **13.93** | **36.15** | **25.53** |
| | Baseline | 15.66 | 19.10 | 24.10 |
| | Untargeted Random KB | 16.02 | 19.54 | 24.51 |
| ReasonIR | Random Entity Augmentation | 17.21 | 20.63 | 24.92 |
| | All Entities | 19.73 | 27.41 | 25.31 |
| | ARGUS (Low-RPS) | **21.13** | **30.60** | **25.53** |

affecting fewer than one entity per document on average, indicating that RPS-guided augmentation concentrates on a relatively small subset of high-risk entities, rather than uniformly expanding the corpus.

We further evaluate ARGUS under constrained augmentation budgets by limiting the number of additional indexed augmented views relative to corpus size. For example, a 20% budget allows at most 20 additional augmented documents per 100 original documents. Table 4 shows that ARGUS yields consistent gains even under small budgets **(20-40%)**, with performance saturating as more high-risk entities are covered.

Finally, we evaluate a strict no-index-growth setting in which original documents are replaced by single LLM-augmented versions instead of indexing additional augmented views. Table 5 shows that ARGUS still improves retrieval quality across benchmarks and retrievers, suggesting that the gains are not solely due to larger indexes, but also improved entity-level accessibility in embedding space.

Together, these results suggest that ARGUS improves dense retrieval robustness through sparse, targeted augmentation without requiring aggressive corpus expansion.

### 7.4. Sensitivity and Ablation Analysis

We next analyze whether ARGUS's gains arise from targeted blind-spot diagnosis rather than generic corpus expansion. We first study sensitivity to the risk threshold $\tau$, which controls the trade-off between augmentation precision and coverage. Lower thresholds focus on only the highest-risk entities, while larger thresholds progressively expand augmentation coverage. Table 6 reports performance across different $\tau$ values. Performance peaks around $\tau \approx 0.3$ and remains stable through $\tau = 0.4$, supporting the robustness of the default threshold used throughout the paper. Larger thresholds eventually yield diminishing returns as augmentation increasingly includes entities that are already sufficiently retrievable.

We additionally evaluate augmentation strategies along two axes: entity grounding (entity-aware vs. untargeted) and en-

tity selection (random, all, or targeted). For each document, let $M$ denote the set of extracted named entities obtained via NER, and let $N \subseteq M$ denote the subset selected for augmentation. Table 7 compares four settings: (1) *Untargeted Random KB*, which inserts $N$ random KB passages independent of document entities, (2) *Random Entity Augmentation*, which augments $N$ randomly selected document entities without using RPS-based selection and augment relevant KB document to them, (3) *All Entities*, which augments $M$ extracted entity in the document, and (4) *ARGUS*, which selectively augments only predicted low-RPS entities.

Untargeted augmentation yields little improvement, suggesting that simply adding external text is insufficient. Random entity augmentation provides moderate gains through entity grounding, but remains weaker than targeted selection. Augmenting all entities further improves coverage, yet still underperforms ARGUS because many already-accessible entities are unnecessarily expanded. In contrast, ARGUS consistently achieves the strongest improvements across benchmarks, indicating that both entity grounding and targeted low-RPS selection are important for remedying embedding-space blind spots.

Taken as a whole, these analyses suggest that ARGUS gains arise not merely from adding text to the index, but from targeted entity-aware augmentation guided by predicted retrievability. Overall, ARGUS provides an effective index-time remedy for retriever blind spots without requiring retriever retraining or query rewriting.

## 8. Conclusion

We studied whether retrieval failures in neural RAG systems arise primarily from sporadic errors or from intrinsic blind spots: systematic failures to retrieve certain entities under practical top-$k$ budgets. To quantify entity-level retrievability, we introduced RPS and a large-scale auditing protocol based on Wikidata-Wikipedia alignment. Our analysis showed that low- and high-RPS entities occupy distinguishable regions in embedding space, enabling lightweight retriever-specific probes to predict blind-spot risk directly from entity embeddings. Building on this signal, we proposed ARGUS, an indexing-time pipeline that diagnoses high-risk entities and remedies them through either KB-guided Document Expansion or LLM Synthesis. Across BRIGHT, IMPLIRET, and RAR-B, ARGUS consistently improves retrieval quality (nDCG@5/10) across diverse neural retrievers without retriever fine-tuning or query rewriting. More broadly, our results suggest that auditing and mitigating blind spots at indexing time is a practical path toward more reliable retrieval for RAG systems. While our study focuses on dense neural retrievers and depends on external KB coverage, we hope this motivates future work on adaptive retriever-aware remedies and hybrid retrieval settings.

## Impact Statement

This paper studies retrieval blind spots in neural information retrieval systems. We further propose RPS for evaluating entity-level retrievability and ARGUS for mitigating these failures to improve the robustness of retrieval systems. Our method does not introduce new ethical concerns beyond those already associated with prior work on improving retrieval systems and retrieval quality.

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

## A. Appendix Overview

This appendix provides additional details and analyses that support the main text. We follow the paper's assess-detect-remedy pipeline: Section 3 (Assess), Section 4 (Detect), Section 5 (ARGUS), and Section 6-7 (Experiments/Results).

## B. Additional Details for Assessing Retriever Blind Spots

This appendix section provides implementation details for the entity-centric audit protocol used to compute RPS and characterize retriever blind spots (Section 3).

### B.1. Entity Sampling and Filtering

**Wikidata sampling.** We begin from a large random sample of Wikidata items and retain those that are linkable to an English Wikipedia page. We denote the filtered set of retained entities by $X$ in the following paragraphs.

**Wikipedia linkage and basic validity checks.** For each candidate entity, we require: (i) a resolvable English Wikipedia page, (ii) a non-empty first paragraph that can be parsed as text, and (iii) a valid surface form (Wikidata label) that can be matched in the paragraph after light normalization. If the valid surface does not appear in the Wikipedia first paragraph, we attach it to the beginning of the Wikipedia first paragraph. We discard malformed entries (e.g., missing pages, empty/very short paragraphs, pages that fail parsing).

**Surface-form grounding.** To reduce noisy alignments, we enforce that the entity's Wikidata surface form appears explicitly in the Wikipedia first paragraph $w_x$. We apply lightweight normalization for matching, including case-folding and whitespace normalization (and, when applicable, punctuation-stripping). If the label occurs multiple times, we keep the earliest occurrence as the mention span $s_x$.

**Neighbor requirement.** Finally, we require each retained entity $x \in X$ to have at least one valid related entity (Section B.2) so that $|\mathcal{T}_x| > 0$ and $\text{RPS}_k(x)$ is well-defined.

### B.2. Wikidata-Wikipedia Alignment and Query Construction Details

**1-hop related entities.** For each target entity $x \in X$, we construct the set of related entities $\mathcal{T}_x$ from 1-hop Wikidata neighbors (entities connected to $x$ by any property). We restrict $\mathcal{T}_x$ to entities that (i) have an English Wikipedia page and (ii) satisfy the same surface-form grounding constraint as targets (their own label appears in their Wikipedia first paragraph, if not, we will add it to the beginning of the text).

**Why we embed queries using paragraphs.** A related entity's label alone may be ambiguous (e.g., polysemous

names). To reduce ambiguity, we form the query representation for each $t \in \mathcal{T}_x$ by encoding $t$'s Wikipedia first paragraph $w_t$ and pooling at the mention span $s_t$ (rather than encoding the short label string in isolation). Concretely, the query embedding for $t$ is $\mathbf{q}_t = g(E_\theta(w_t), s_t)$, matching the same representation format used for candidates (Section 3.2).

**Controlling query sets.** We treat each $t \in \mathcal{T}_x$ as a proxy query context in which $x$ should be retrievable. In practice, $\mathcal{T}_x$ can vary in size across entities; RPS averages hits across all available related entities for each $x$ (Section 3.2).

### B.3. Neutral Pool Construction and Disjointness Checks

**Motivation.** RPS is designed to measure whether an entity is retrieved due to genuine geometric alignment rather than random collisions. We therefore evaluate each target entity under *controlled competition* against a neutral pool that is (by construction) unrelated to the query entity.

**Neutral candidate eligibility.** Each neutral candidate $z$ must: (i) have an English Wikipedia page with a valid first paragraph $w_z$, (ii) satisfy surface-form grounding (its label appears in $w_z$; if not, we concatenate it to the beginning of the first paragraph), and (iii) pass the KG-disjointness constraint described below.

**KG-disjointness constraint.** For a related entity (query) $t$, we define $\mathrm{Nbr}(t)$ as its set of 1-hop Wikidata neighbors (entities directly connected to $t$ by any property). We require each neutral $z \in \mathcal{Z}_{\mathrm{neut}}(t)$ to be *not directly connected* to $t$ in the Wikidata graph, i.e.,

$$z \notin \mathrm{Nbr}(t).$$

(equivalently, no 1-hop KG edge exists between $z$ and $t$) This constraint reduces the chance that a "neutral" candidate is trivially related to $t$ via an explicit KG link, making the neutral pool a stronger control.

**Per-query neutral pools and sampling.** We maintain a related-entity-specific neutral pool $\mathcal{D}_{\mathrm{neut}}(t)$ for each query entity $t$. For each retrieval trial, we sample $N - 1$ neutrals uniformly from $\mathcal{D}_{\mathrm{neut}}(t)$ and evaluate whether the target $x$ appears in the top-$k$ among the $N$ candidates (Section 3.2).

**Pool size.** Unless otherwise stated, we use $N = 800$ neutrals for the audit, motivated by the stability analysis in Figure 3. We further analyze the sensitivity to the user-chosen retrieval window $k$ at fixed $N = 800$ in Figure 5.

### B.4. RPS Implementation Details

**Retriever-specific representations.** Different dense retrievers expose different embedding interfaces (e.g., sentence-level vectors vs. token-level hidden states). We unify them through a retriever-specific pooling function $g(\cdot)$ that returns

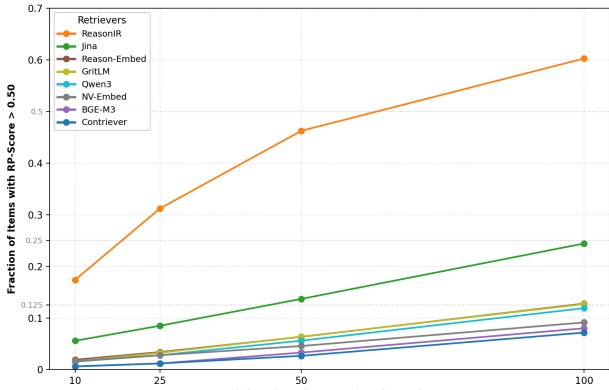

*Figure 5.* **Sensitivity of retrieval consistency to the retrieval window size ($k$) at fixed $N = 800$.** Increasing the user-defined parameter $k$ expands the retrieval scope. Standard retrievers (e.g., Contriever, BGE-M3) exhibit approximately linear growth consistent with statistical scaling of the random-hit window. In contrast, ReasonIR displays a non-linear trajectory with a mild elbow, indicating that its gains are driven by learned geometric structure rather than simple expansion of candidate slots.

a single $h$-dimensional vector per entity instance.

**Mention span extraction.** Given a paragraph $w$ and an entity label, we identify a token span $s$ corresponding to the first grounded occurrence of the label in $w$ (after the normalization described in Appendix B.1). If the label tokenizes into multiple subwords, $s$ covers the full subword span.

**Pooling operator $g(\cdot)$.** Let $H = E_\theta(w)$ denote the representation produced by the retriever encoder for $w$. We use:

- **Span pooling (token-level models).** If $H$ provides token-level embeddings, we compute the entity embedding by averaging token representations over the mention span:

$$g(H, s) = \frac{1}{|s|} \sum_{i \in s} H_i.$$

- **Sentence-level models.** If the retriever only returns a single vector for the whole input, we set $g(H, s)$ to that vector (the span is ignored but the same paragraph-level input is used for all entities).

This definition ensures that the *same* procedure is applied to targets, queries (related entities), and neutrals: each is represented by its Wikipedia first paragraph, with mention-aware pooling when available.

**Similarity and ranking.** We rank candidates by cosine similarity $\cos(\mathbf{q}_t, \mathbf{e}_c)$ and define $\mathrm{Hit}(x, t)$ using a top-$k$ cutoff (Section 3.2). For completeness, when retrievers

support optional embedding normalization, we follow the retriever's recommended inference-time practice and then apply cosine similarity consistently across models.

### B.5. Additional Geometry Visualizations

This subsection provides the complete set of 2D LDA projections for all evaluated retrievers, extending the representative main-text visualization (Figure 2) in 6. For each retriever, we compute LDA on entity embeddings after labeling entities into RPS terciles (low/mid/high), and we visualize how separable regions evolve under increasing neutral pool sizes $N$ (with $k = 50$). These plots support the central observation that blind spots correspond to structured regions in representation space.

### B.6. Sensitivity to $k$ and Other Audit Hyperparameters

RPS is defined with respect to the user-selected retrieval budget $k$, so it is important to understand how audit conclusions change as $k$ varies. At fixed neutral pool size $N = 800$, increasing $k$ mechanically increases the probability of a hit under random ranking (chance baseline $\approx k/N$), and correspondingly increases the fraction of entities with $\text{RPS}_k > 0.5$. In our experiments, standard retrievers tend to exhibit near-linear scaling with $k$, consistent with expanding the candidate window, whereas more robust retrievers exhibit departures from purely linear behavior, indicating that improvements are driven by learned geometric structure rather than chance alone.

## C. Additional Details for Detecting Blind Spots

### C.1. Probe Families and Hyperparameter Sweep

This section provides additional implementation details for the embedding-based diagnostic probes introduced in Section 4. Our goal is to learn a retriever-specific predictor $h_\phi : \mathbb{R}^d \to [0, 1]$ that maps an entity embedding $\mathbf{e}_x$ to $\widehat{\text{RPS}}_k(x)$, enabling *pre-index* risk estimation directly from representation geometry.

**Input representation.** For each retriever, we use the same entity embedding construction as in Section 3.2: the input vector $\mathbf{e}_x = g(E_\theta(w_x), s_x)$ is a mention-pooled embedding extracted from the target retriever encoder $E_\theta$ over the entity context $w_x$ with span $s_x$. Unless stated otherwise, probes operate on the raw embedding vector; we do not require query-conditioned features or retrieval simulations.

**Probe families.** We evaluate three probe families that span linear, non-linear, and tree-based function classes: (i) **Linear probes** (Ridge regression) as a strong, low-variance baseline, (ii) **MLP probes** to capture non-linear structure in embedding geometry with layers of dense neural networks,

and (iii) **Gradient-boosted trees** (XGBOOST) as a flexible non-linear model well-suited to tabular features. All probes are trained *separately per retriever*, since embedding spaces and pooling operators differ across architectures.

**Training splits and objective.** For each retriever, we create a standard train/validation/test split over entities, and train probes to regress to empirical $\text{RPS}_k(x)$ computed under our stable audit setting ($N=800$ neutrals, fixed $k$). Hyperparameters are selected by minimizing validation RMSE; final metrics are reported on the held-out test split (Table 1). We also report a semi-classification view by discretizing entities into three RPS bands (*low/mid/high*) and evaluating accuracy and macro-F1 using the same predicted scores (Section 4.2). For completeness, the full sweep results across all probe families and hyperparameter settings are provided in Table 8.

**Hyperparameter sweep.** For completeness, we summarize the principal hyperparameters explored for each family. Unless stated otherwise, all sweeps are performed independently per retriever and per $k$.

- **Ridge regression.** Regularization strength $\alpha \in \{10^{-6}, 10^{-5}, \dots, 10^3\}$ (log-spaced); intercept enabled; features unnormalized or standardized (both evaluated).

- **MLP.** Hidden widths $\in \{256, 512, 1024\}$; depth $\in \{1, 2, 3\}$; dropout $\in \{0.0, 0.1, 0.2\}$; learning rate $\in \{10^{-4}, 3\times10^{-4}, 10^{-3}\}$; batch size $\in \{256, 512, 1024\}$; early stopping on validation RMSE with patience 10.

- **XGBOOST.** Number of trees $\in \{300, 600, 1000\}$; max depth $\in \{4, 6, 8\}$; learning rate $\eta \in \{0.03, 0.05, 0.1\}$; subsample $\in \{0.7, 0.9, 1.0\}$; column subsample $\in \{0.7, 0.9, 1.0\}$; minimum child weight $\in \{1, 5, 10\}$; L2 regularization $\lambda \in \{0, 1, 10\}$. We use early stopping on validation RMSE.

**Model selection.** For each retriever, we select the probe that attains the lowest validation RMSE and report its test performance. In our experiments, XGBOOST is frequently the best-performing family, although linear and MLP probes can be competitive depending on the retriever and pooling scheme. We emphasize that probe performance is not the primary contribution; rather, strong performance across families supports the conclusion that retrievability risk is encoded in embedding geometry and can be detected pre-index.

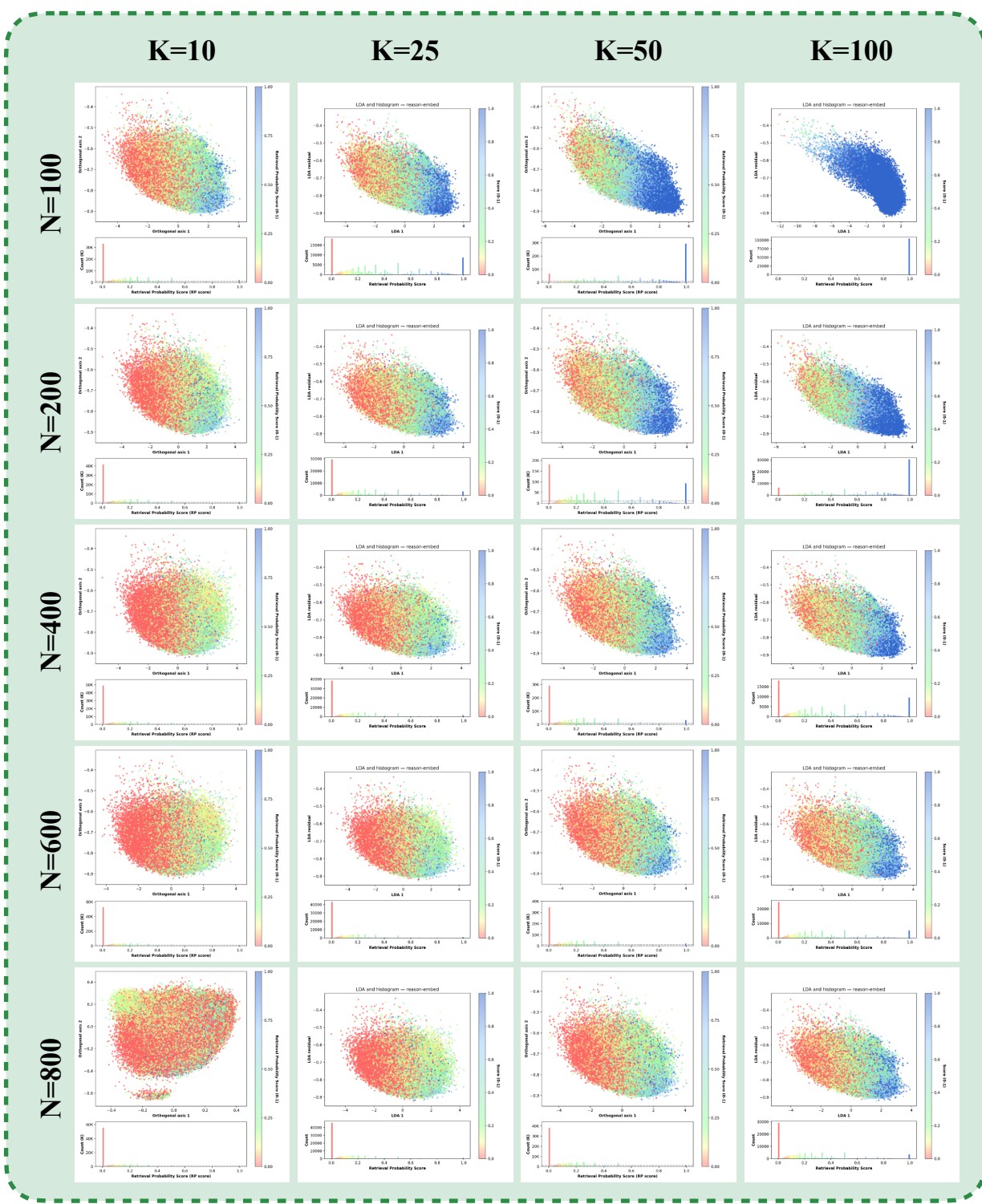

*Figure 6.* **Extended geometric visualization of two-dimensional LDA projections for all evaluated retrievers under increasing neutral pool sizes** $N$. Consistent with the main analysis, entities are labeled by RPS terciles (low/mid/high) at $k = 50$. The full benchmark reveals that standard dense retrievers (e.g., BGE-M3, Qwen3, GritLM) exhibit a collapsing geometric structure similar to Contriever, where low-RPS regions dominate as competition increases. In contrast, specialized models like Jina and ReasonIR maintain more distinct high-RPS clusters, though intrinsic blind spots persist across all architectures.

*Table 8.* **Comprehensive performance benchmark of embedding-based diagnostic probes across diverse architectures**. We compare learned probes (Ridge, XGBoost, MLP) against trivial baselines (All-One, All-Zero) for predicting RPS ($k = 50$, $N = 800$). Across all retrievers, learned probes consistently achieve significantly lower error (RMSE) and higher correlation than baselines, validating that blind spots are predictable geometric properties rather than random noise. The optimal configuration for each retriever (selected via lowest RMSE) is summarized in the main text (1).

| Retriever | Architecture | Regression Metrics | | | | Semi-Classification Metrics | | | | | |
| | | RMSE ($\downarrow$) | MAE ($\downarrow$) | Pearson $r$ ($\uparrow$) | Spearman $\rho$ ($\uparrow$) | Macro-F1 ($\uparrow$) | Macro-Rec. ($\uparrow$) | Macro-Prec. ($\uparrow$) | Prec$_{weighted}$ ($\uparrow$) | F1$_{weighted}$ ($\uparrow$) | Accuracy ($\uparrow$) |
|---|---|---|---|---|---|---|---|---|---|---|---|
| **BGE-M3** | All One | 0.783 | 0.749 | 0.000 | 0.000 | 0.027 | 0.333 | 0.049 | 0.080 | 0.012 | 0.080 |
| | All Zero | 0.340 | 0.251 | 0.000 | 0.000 | 0.242 | 0.333 | 0.280 | 0.726 | 0.610 | 0.726 |
| | Ridge | 0.169 | 0.121 | 0.674 | 0.638 | 0.655 | 0.528 | 0.548 | 0.767 | 0.754 | 0.767 |
| | XGBoost | **0.168** | **0.118** | **0.681** | **0.644** | **0.658** | **0.540** | **0.573** | **0.781** | **0.762** | **0.781** |
| | MLP | 0.170 | 0.122 | 0.671 | 0.632 | 0.654 | 0.522 | 0.542 | 0.766 | 0.752 | 0.766 |
| **CONTRIEVER** | All One | 0.798 | 0.770 | 0.000 | 0.000 | 0.019 | 0.333 | 0.036 | 0.058 | 0.006 | 0.058 |
| | All Zero | 0.310 | 0.230 | 0.000 | 0.000 | 0.251 | 0.333 | 0.286 | 0.753 | 0.647 | 0.753 |
| | Ridge | 0.168 | 0.121 | 0.591 | 0.582 | **0.774** | 0.476 | 0.460 | 0.778 | 0.759 | 0.778 |
| | XGBoost | **0.157** | **0.109** | **0.658** | **0.622** | 0.727 | **0.501** | **0.506** | **0.795** | **0.777** | **0.795** |
| | MLP | 0.168 | 0.121 | 0.588 | 0.579 | **0.774** | 0.475 | 0.460 | 0.779 | 0.759 | 0.779 |
| **QWEN3-EMBEDDING** | All One | 0.760 | 0.719 | 0.000 | 0.000 | 0.036 | 0.333 | 0.065 | 0.108 | 0.021 | 0.108 |
| | All Zero | 0.373 | 0.281 | 0.000 | 0.000 | 0.226 | 0.333 | 0.269 | 0.677 | 0.547 | 0.677 |
| | Ridge | 0.177 | 0.135 | 0.699 | 0.647 | 0.677 | 0.582 | 0.593 | 0.721 | 0.727 | 0.721 |
| | XGBoost | **0.153** | **0.111** | **0.781** | **0.721** | **0.699** | **0.619** | **0.646** | **0.764** | **0.760** | **0.764** |
| | MLP | 0.180 | 0.137 | 0.694 | 0.637 | 0.670 | 0.585 | 0.597 | 0.720 | 0.726 | 0.720 |
| **GRITLM-7B** | All One | 0.753 | 0.716 | 0.000 | 0.000 | 0.036 | 0.333 | 0.065 | 0.108 | 0.021 | 0.108 |
| | All Zero | 0.369 | 0.284 | 0.000 | 0.000 | 0.224 | 0.333 | 0.268 | 0.673 | 0.542 | 0.673 |
| | Ridge | 0.171 | 0.128 | 0.696 | 0.638 | 0.658 | 0.576 | 0.589 | 0.734 | 0.733 | 0.734 |
| | XGBoost | **0.157** | **0.115** | **0.745** | **0.677** | **0.682** | **0.595** | **0.620** | **0.762** | **0.754** | **0.762** |
| | MLP | 0.162 | 0.116 | 0.731 | 0.669 | 0.671 | 0.586 | 0.611 | 0.760 | 0.750 | 0.760 |
| **REASON-EMBED** | All One | 0.750 | 0.710 | 0.000 | 0.000 | 0.036 | 0.333 | 0.066 | 0.109 | 0.022 | 0.109 |
| | All Zero | 0.377 | 0.290 | 0.000 | 0.000 | 0.220 | 0.333 | 0.265 | 0.659 | 0.523 | 0.659 |
| | Ridge | 0.169 | 0.128 | 0.716 | 0.693 | 0.680 | 0.577 | 0.576 | 0.732 | 0.729 | 0.732 |
| | XGBoost | **0.156** | **0.114** | **0.764** | **0.742** | **0.688** | **0.595** | **0.609** | **0.752** | **0.748** | **0.752** |
| | MLP | 0.169 | 0.129 | 0.716 | 0.694 | 0.680 | 0.580 | 0.578 | 0.733 | 0.731 | 0.733 |
| **NV-EMBED-V2** | All One | 0.772 | 0.738 | 0.000 | 0.000 | 0.028 | 0.333 | 0.052 | 0.085 | 0.013 | 0.085 |
| | All Zero | 0.345 | 0.262 | 0.000 | 0.000 | 0.239 | 0.333 | 0.279 | 0.718 | 0.601 | 0.718 |
| | Ridge | 0.179 | 0.132 | 0.624 | 0.576 | 0.626 | **0.535** | **0.550** | 0.736 | 0.734 | 0.736 |
| | XGBoost | **0.173** | **0.124** | **0.640** | **0.595** | **0.657** | 0.517 | 0.541 | **0.757** | **0.740** | **0.757** |
| | MLP | 0.180 | 0.133 | 0.623 | 0.576 | 0.627 | **0.535** | 0.549 | 0.736 | 0.734 | 0.736 |
| **JINA-V3** | All One | 0.703 | 0.661 | 0.000 | 0.000 | 0.041 | 0.333 | 0.073 | 0.124 | 0.027 | 0.124 |
| | All Zero | 0.414 | 0.339 | 0.000 | 0.000 | 0.183 | 0.333 | 0.236 | 0.548 | 0.388 | 0.548 |
| | Ridge | **0.178** | **0.137** | **0.667** | **0.659** | **0.641** | 0.557 | 0.574 | 0.655 | 0.653 | 0.655 |
| | XGBoost | 0.179 | 0.138 | 0.661 | 0.650 | 0.633 | 0.537 | 0.551 | 0.650 | 0.643 | 0.650 |
| | MLP | **0.178** | **0.137** | **0.667** | 0.658 | **0.641** | 0.558 | 0.575 | 0.656 | 0.654 | 0.656 |
| **REASONIR-8B** | All One | 0.558 | 0.500 | 0.000 | 0.000 | 0.099 | 0.333 | 0.153 | 0.297 | 0.136 | 0.297 |
| | All Zero | 0.558 | 0.500 | 0.000 | 0.000 | 0.091 | 0.333 | 0.143 | 0.274 | 0.118 | 0.274 |
| | Ridge | **0.156** | **0.121** | **0.779** | **0.788** | **0.710** | **0.658** | **0.674** | **0.674** | **0.674** | **0.674** |
| | XGBoost | **0.156** | **0.121** | 0.776 | 0.781 | 0.707 | 0.641 | 0.659 | 0.662 | 0.661 | 0.662 |
| | MLP | **0.156** | **0.121** | 0.778 | 0.787 | 0.708 | 0.657 | 0.673 | 0.672 | 0.673 | 0.672 |

## C.2. Calibration and Residual Diagnostics

To complement aggregate regression/classification metrics, we analyze whether probes are *well-calibrated* and whether their errors show systematic bias. Figure 7 reports two diagnostics for the best probe per retriever.

**Predicted-empirical density.** The top row visualizes the joint density of predicted $\widehat{\text{RPS}}_k$ versus empirical $\text{RPS}_k$. Concentration along the diagonal indicates good calibration, while off-diagonal mass reveals over- or under-estimation regimes. For standard retrievers, the density is heavily concentrated at low empirical RPS, reflecting that a large fraction of entities fall into low-retrievability regions under our stringent neutral competition setting ($N=800$). Reasoning-oriented retrievers show a broader spread with higher empirical RPS mass, consistent with their stronger average retrievability.

**Residual distribution.** The bottom row plots residuals ($\text{RPS}_k - \widehat{\text{RPS}}_k$). Across retrievers, residuals are centered near zero with limited skew, indicating that the probes do not exhibit large systematic optimism or pessimism. The remaining dispersion is primarily attributable to (i) retriever-specific noise in empirical RPS estimation due to finite query sets $|\mathcal{T}_x|$, and (ii) class imbalance in the underlying RPS distribution (many low-RPS entities for standard retrievers). Overall, these diagnostics support the use of probe predictions as practical, *pre-index* risk scores for threshold-based flagging in ARGUS.

## D. Additional ARGUS Implementation Details

This appendix summarizes practical details for implementing ARGUS (Section 5), including named-entity extraction, handling repeated entities, and the two augmentation modes used in our experiments.

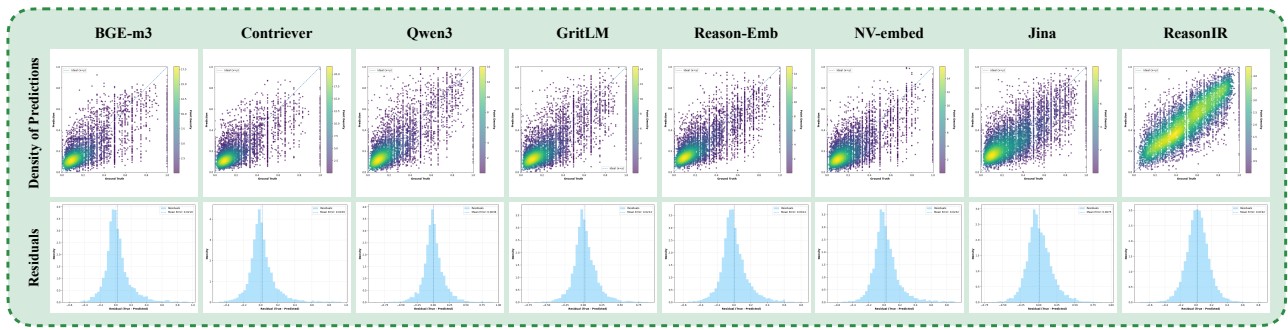

*Figure 7.* **Calibration analysis of embedding-based diagnostic probes (Predicted vs. Empirical RPS). (Top) Prediction density:** Heatmaps of predicted versus true RPS illustrate that probes recover the overall retrievability structure. For standard retrievers (e.g., Contriever), the concentration near low RPS reflects the skew toward geometrically hard-to-retrieve entities (blind spots). In contrast, ReasonIR shows a more dispersed mass consistent with higher entity retrievability, which the probe tracks. **(Bottom) Residual analysis:** Distributions of residuals (True − Predicted) are centered near zero, indicating limited systematic over/under-estimation and supporting the use of these probes for preindex quantification of retrievability.

### D.1. Named Entity Extraction

**NER model and outputs.** We extract named entities from each corpus document $D$ using an off-the-shelf NER tagger, `dslim/bert-base-NER`. We use *named entity* to denote an NER-extracted span in $D$, represented as a tuple $(m, s)$ where $m$ is the extracted surface form (string) and $s$ is its character span (start/end offsets) in $D$.

**Span handling and mapping to retriever tokens.** Because retriever encoders operate on tokenized inputs, we map each character span $s$ to the corresponding token indices under the target retriever's tokenizer. If a span aligns to multiple wordpieces, we treat the entire aligned token range as the entity span for pooling.

### D.2. Repeated Mentions and Document-Level Risk Aggregation

A document may contain multiple mentions of the same named entity surface form (or closely related surface forms). ARGUS uses a conservative aggregation scheme to avoid missing high-risk cases while preventing redundant augmentation.

**Per-mention scoring.** For each extracted mention $(m, s)$ in document $D$, we compute a context-dependent embedding using the target retriever and the same span pooling operator as in Section 3.2:

$$\mathbf{e}_{m,D,s} = g(E_\theta(D), s).$$

We then interpret the mention's retrievability under a top-$k$ budget as the expected top-$k$ hit probability over its related-query set $\mathcal{T}_m$:

$$\widehat{\mathrm{RPS}}_k(m \mid D, s) = \mathbb{E}_{t \sim \mathrm{Uniform}(\mathcal{T}_m)}[\mathrm{Hit}_k(m, t)],$$

where $\mathrm{Hit}_k(m, t) = \mathbb{I}[\mathrm{rank}(m \mid t) \leq k]$ is computed by ranking $m$ against the related-entity-specific neutral pool for $t$ (as in Section 3.2). In practice, we estimate this expectation by the empirical mean of hit indicators over $t \in \mathcal{T}_m$.

**Risk-conserving aggregation for repeated entities.** If the same surface form $m$ appears multiple times in $D$ with spans $\{s_1, \ldots, s_k\}$, we assign the entity a single document-level risk score using the minimum predicted score:

$$\widehat{\mathrm{RPS}}_k(m \mid D) = \min_j \widehat{\mathrm{RPS}}_k(m \mid D, s_j).$$

This *risk-conserving* rule ensures that if any occurrence of $e$ is embedded into a low-retrievability region (e.g., due to local context), the entity is treated as high-risk.

**De-duplication and augmentation once per entity.** We then flag $e$ as high-risk if $\widehat{\mathrm{RPS}}_k(e \mid D) < \tau$ and include it in $\mathcal{E}_{\mathrm{risk}}(D)$ (Section 5.1). Importantly, we apply augmentation *once per flagged entity per document*, even if the entity appears repeatedly:

$$\mathcal{E}_{\mathrm{risk}}(D) = \{ e : \widehat{\mathrm{RPS}}_k(e \mid D) < \tau \}.$$

This avoids multiplying near-duplicate augmented views solely due to repeated mentions while preserving the conservative detection behavior via the minimum rule above.

### D.3. ARGUS Remedy Procedure (Non-Code Summary)

This subsection summarizes the remedy stage at indexing time (Section 5.2) in three steps.

**Step 1: Reference KB retrieval.** For each flagged entity $m \in \mathcal{E}_{\mathrm{risk}}(D)$, we retrieve a concise defining context from a Reference KB (Wikipedia first paragraphs in our experiments). We query the KB using the entity surface form $m$

(not the full document) and retrieve the top $k_{\text{int}}$ passages with BM25s ($k_{\text{int}}=2$):

$$\{p_{m,1}, \ldots, p_{m,k_{\text{Aug}}}\} \leftarrow \text{BM25s}(\text{KB}, \text{query} = m).$$

**Step 2: Construct augmented document views.** We instantiate two alternatives:

**(i) Document Expansion (Concatenation).** For each retrieved passage $p_{m,i}$, we create an expanded view by appending the passage to the document:

$$D_{m,i}^{\text{exp}} = D \parallel p_{m,i}.$$

If $N_D = |\mathcal{E}_{\text{risk}}(D)|$, this produces $k_{\text{int}} \cdot N_D$ expanded views, in addition to indexing the original document.

**(ii) KB-guided LLM Synthesis.** We aggregate all retrieved passages for all flagged entities in $D$ and generate a single unified augmented view:

$$D^{\text{synth}} = \text{LLM}(D, \{p_{e,i}\}_{e \in \mathcal{E}_{\text{risk}}(D), \, i \leq k_{\text{int}}}),$$

where the LLM inserts short, entity-focused clarifications only where needed (prompt details below). This option produces exactly one synthesized view per document.

**Step 3: Indexing strategy.** In both modes, we index augmented views *alongside* the original document (never replacing it). Thus, the index contains: (i) the original $D$, (ii) all $D_{e,i}^{\text{exp}}$ views (expansion), or (iii) the single $D^{\text{synth}}$ view (synthesis). Retrieval is then performed over the expanded index using the same target retriever and scoring procedure as in the **Original** baseline.

### D.4. Prompt Template for KB-guided LLM Synthesis

Figure 8 shows the prompt template used for *KB-guided LLM Synthesis*. Given document $D$ and the retrieved KB passages for its flagged entities, the prompt instructs the model to produce a single augmented document that preserves the original meaning and adds only minimal clarifications needed to improve retrievability.

## E. Additional Analyses

### E.1. Association Between Entity-Level RPS and Retrieval Success

This analysis probes whether entity-level retrievability (as measured by RPS) is *associated* with downstream document retrieval success in benchmark settings. We emphasize that retrieval outcomes depend on many query–document factors (e.g., query phrasing, document length, topicality, and semantic match), so this analysis is not intended to establish causality.

**Setup.** For each benchmark instance, we consider the gold (relevant) document(s) and extract named entities using the same NER procedure as in Appendix D.1. For each extracted entity, we estimate its retrievability score under the target retriever using our diagnostic probe, yielding $\widehat{\text{RPS}}_k(e \mid D)$. We then separate gold documents into two groups: those that are successfully retrieved within the top-$k$ window and those that are not.

**Statistic.** For each group, we compute the maximum predicted RPS among entities contained in the gold document:

$$\max_{e \in \mathcal{E}(D)} \widehat{\text{RPS}}_k(e \mid D),$$

and report the difference in this maximum between the retrieved and unretrieved groups. A positive delta indicates that retrieved gold documents tend to contain at least one entity with higher geometric visibility, suggesting an association between entity-level retrievability and retrieval success in entity-centric settings.

**Results and interpretation.** Table 9 summarizes this difference across BRIGHT and RAR-b subsets. Positive deltas (shown in green) indicate subsets where retrieved documents tend to contain entities with higher RPS, while negative deltas indicate subsets where this association is weaker or absent—consistent with retrieval being influenced by additional query–document factors beyond entity geometry. Overall, these results support the view that RPS captures a meaningful component of retrieval difficulty in entity-centric scenarios, while also highlighting that it is not the sole determinant of retrieval success.

## F. Additional Experimental Results

This appendix provides additional experimental results beyond the subset displayed in the main paper for readability. We include (i) full per-task tables for each benchmark and (ii) extended cutoff metrics to complement the main nDCG@5/10 reporting.

### F.1. Full Per-Task Tables and Extended Cutoffs

**Full per-task results.** We report complete per-task breakdowns for all benchmarks evaluated in Table 2, including: **BRIGHT** (all 10 domains), **ImpliRet** (both settings), and **RAR-b** (all 7 subsets). These tables mirror the main-table format, comparing **Original** indexing against ARGUS with *Document Expansion* and *LLM Synthesis* for each retriever.

**Extended cutoff metrics.** In addition to nDCG@5/10 (main paper), we report nDCG at larger cutoffs to characterize broader recall-oriented behavior:

$$\text{nDCG@20}, \ \text{nDCG@50}.$$

You are an expert editor and fact-checker. Your goal is to improve a text for a search engine by adding specific descriptions, but ONLY where the Wikipedia context clearly matches the meaning in the text.

### Task Instructions:
1. Read the 'Input Text'.
2. Review the list of 'Candidate Entities' and their provided Wikipedia context.
3. For EACH entity, perform a Context Check:
    - Does the entity in the 'Input Text' refer to the exact same concept described in the Wikipedia context?
    - Example of MISMATCH: Input says 'it was half past nine' (time), but Wikipedia describes 'Half Past Nine' (the album). -> ACTION: Do NOT augment.
    - Example of MATCH: Input says 'St. Peter', Wikipedia describes 'St. Peter Church in Zurich'. -> ACTION: AUGMENT.
4. Generation Step: Rewrite the Input Text. If (and ONLY if) an entity passes the Context Check, insert a short description (max 5 words, enclosed in commas) immediately after the entity.
5. If an entity fails the check or you are uncertain, leave it exactly as it is.
6. The short description must be a concise type/role/category derived from the Wikipedia context (e.g., 'industrial EBM album', 'Swiss Catholic church in Zurich'), not a long story or extra sentence.
7. Do not change the wording, order, or punctuation of the Input Text, except for inserting these short descriptions. Do not introduce new entities or facts that are not supported by the Wikipedia context.
8. If the same entity appears multiple times in the Input Text, augment only the first occurrence and leave the others unchanged.
9. Your entire output MUST be only the fully augmented text, with no headings, labels, explanations, or surrounding formatting.

### Candidate Entities & Context: [ENTITY_1] :
[WIKI_FIRST_PARAGRAPH_1] [ENTITY_2] : [WIKI_FIRST_PARAGRAPH_2] …

### Input Text:
[ORIGINAL_DOCUMENT_TEXT]

### Final Annotated Text:

*Figure 8.* **Prompt template used for KB-guided LLM synthesis in ARGUS**. Candidate entities are paired with retrieved Wikipedia first-paragraph contexts; the model inserts short comma-delimited descriptors only when the context check passes.

Where space permits, we additionally include Recall@k at matching cutoffs to highlight changes in coverage at larger retrieval windows. These extended-cutoff results are consistent with the main findings: ARGUS improves retrieval quality across a wide range of retrievers and tasks, with *Document Expansion* typically providing the most stable gains and *LLM Synthesis* offering a more index-efficient alternative with greater retriever-dependent variance.

### F.2. Additional Notes on Metrics and Aggregation

**Metric definitions.** We use nDCG@k as the primary metric to capture ranked relevance quality under practical top-$k$ budgets. For each benchmark, we compute nDCG@k following the dataset's standard evaluation protocol and relevance labeling.

**Full Benchmark Avg.** To summarize suite-level performance without relying on a small set of displayed tasks, we report *Full Benchmark Avg.* for each benchmark as the arithmetic mean of the metric across the full evaluated task suite:

$$\text{FullAvg} = \frac{1}{|\mathcal{S}|} \sum_{s \in \mathcal{S}} \text{Metric}(s),$$

where $\mathcal{S}$ is the set of tasks/subsets for that benchmark (BRIGHT: 10 domains; ImpliRet: 2 settings; RAR-b: 7 subsets). This aggregation is computed separately for each retriever and each system configuration (Original, ARGUS-Expansion, ARGUS-Synthesis).

**Interpretation.** Because Full Benchmark Avg. averages across all tasks, it provides a robustness-oriented summary of performance and reduces sensitivity to which per-task columns are displayed in the main paper. Extended per-task results and additional cutoffs (e.g., nDCG@20/50) are reported in Table 10.

*Table 9.* **Difference in maximum RPS between retrieved and unretrieved gold documents (BRIGHT, RAR-B).** Positive deltas (green) indicate that retrieved documents tend to contain entities with higher geometric visibility (RPS), suggesting an association between entity-level retrievability and retrieval success in entity-centric settings. Negative deltas indicate subsets where this association is weaker or absent, consistent with retrieval being influenced by additional query–document factors beyond entity geometry.

| Retriever | k | BRIGHT | | | | RAR-b | |
|---|---|---|---|---|---|---|---|
| | | Biology | Psychology | Theorems | Questions | HellaSwag | PIQA |
| BGE-M3 | 10 | 35.00/31.56(+3.45) | 35.32/33.31(+2.01) | 24.72/22.89(+1.83) | 22.96/25.91(-2.95) | 25.97/24.50(+1.47) | 20.33/28.56(-8.23) |
| | 20 | 36.10/30.73(+5.37) | 35.34/32.76(+2.58) | 23.38/23.07(+0.31) | 25.42/25.45(-0.04) | 25.36/25.67(-0.30) | 20.24/30.36(-10.12) |
| | 50 | 34.97/30.55(+4.43) | 34.67/33.74(+0.93) | 21.71/23.53(-1.82) | 24.90/25.64(-0.74) | 25.55/24.99(+0.56) | 20.93/31.08(-10.15) |
| CONTRIEVER | 10 | 33.55/30.04(+3.52) | 31.20/28.48(+2.72) | 44.60/20.68(+23.93) | 32.10/24.50(+7.60) | 23.51/18.31(+5.19) | 19.52/18.30(+1.22) |
| | 20 | 32.16/30.33(+1.83) | 31.00/28.52(+2.49) | 36.60/20.52(+16.09) | 32.57/24.13(+8.44) | 23.00/17.97(+5.02) | 19.27/18.88(+0.39) |
| | 50 | 31.43/30.77(+0.66) | 30.79/27.63(+3.16) | 37.93/20.01(+17.92) | 32.21/23.81(+8.41) | 22.59/18.37(+4.22) | 19.27/18.88(+0.39) |
| textscEeason-Embed | 10 | 46.40/47.81(-1.41) | 52.62/47.03(+5.60) | 34.95/36.67(-1.72) | 37.52/33.09(+4.43) | 37.57/40.53(-2.96) | 48.74/31.93(+16.81) |
| | 20 | 46.55/47.87(-1.33) | 52.34/46.96(+5.38) | 33.26/37.98(-4.72) | 37.52/33.09(+4.43) | 38.12/39.99(-1.88) | 47.81/31.61(+16.20) |
| | 50 | 46.91/47.36(-0.45) | 49.83/49.20(+0.63) | 34.15/37.93(-3.78) | 37.28/32.97(+4.31) | 39.35/35.59(+3.76) | 43.08/33.20(+9.89) |
| GritLM-7B | 10 | 46.80/42.68(+4.13) | 35.69/39.20(-3.51) | 30.21/26.91(+3.30) | 24.48/25.55(-1.07) | 28.98/26.15(+2.84) | 24.80/26.14(-1.34) |
| | 20 | 47.53/42.33(+5.20) | 35.32/39.53(-4.21) | 30.21/26.91(+3.30) | 26.15/25.35(+0.80) | 28.15/26.82(+1.33) | 24.80/26.14(-1.34) |
| | 50 | 46.72/42.26(+4.46) | 37.14/39.73(-2.58) | 30.21/26.91(+3.30) | 27.19/25.13(+2.06) | 27.24/28.27(-1.03) | 24.80/26.14(-1.34) |
| Jina | 10 | 68.52/66.33(+2.19) | 67.76/68.34(-0.58) | 65.97/54.71(+11.26) | 55.59/49.69(+5.90) | 67.22/64.35(+2.87) | 53.43/62.42(-8.99) |
| | 20 | 67.87/66.75(+1.12) | 67.86/68.26(-0.39) | 65.97/54.71(+11.26) | 51.97/51.55(+0.42) | 67.17/64.11(+3.06) | 54.84/59.26(-4.42) |
| | 50 | 66.51/69.48(-2.98) | 68.05/68.10(-0.04) | 59.39/56.21(+3.19) | 54.02/49.63(+4.39) | 67.28/61.75(+5.53) | 54.84/59.26(-4.42) |

*Table 10.* **Complete downstream retrieval performance (nDCG@5/10/20/50) across all individual tasks in BRIGHT, IMPLIRET, and RAR-B.** This table complements Table 2 (main text) by providing the granular performance breakdown for every specific sub-domain (e.g., all 10 subject categories in BRIGHT, all 7 tasks in RAR-B). We observe that ARGUS (via Document Expansion or LLM Synthesis) yields consistent improvements across the vast majority of individual tasks, confirming that the holistic gains reported in the main paper are driven by robust, widespread enhancements rather than outlier performance in a few categories. **Colors**: **Best Result**, 2nd Best, 3rd Best.

