# OpenReview forum: "With Argus Eyes: Assessing Retrieval Gaps via Uncertainty Scoring to Detect and Remedy Retrieval Blind Spots"
_ICML.cc/2026/Conference — ICML 2026 regular_

### Official Review · Reviewer_ULoR · 2026-03-11

**Soundness:** 4
**Presentation:** 3
**Significance:** 3
**Originality:** 4
**Overall Recommendation:** 5
**Confidence:** 3

**Summary:**

Overall, the submission studies the concept of entity-level retrievability in neural retrieval systems for RAG pipelines. The authors identify that existing retrieval evaluation methods are query-centric and post-hoc, limiting their ability to preemptively identify systematic retrieval failures. To address this, they introduce the Retrieval Probability Score (RPS), which quantifies an entity's likelihood of being retrieved across relevant queries under a fixed top-k budget. The paper demonstrates that RPS is predictable from embedding geometry and proposes ARGUS, a diagnosis-to-remedy pipeline that flags high-risk entities and augments documents with targeted knowledge base context. This is a solid contribution to retrieval-augmented generation that addresses a real limitation in current neural retrieval systems. The core insight that entity-level retrievability risk is geometrically encoded and predictable is novel and well-validated. However, the paper's accessibility could be significantly improved with clearer exposition of foundational concepts and better figure presentation. With revisions addressing these clarity issues, this would be a strong accept.

**Compliance With Llm Reviewing Policy:**

Affirmed.

**Final Justification:**

Thanks for the response. The authors' rebuttal effectively addressed the raised issues. As the core strengths of the paper remain well-supported, I am pleased to maintain my positive assessment.

**Key Questions For Authors:**

1. In Table 2, LLM Synthesis sometimes underperforms Document Expansion (e.g., JINA-V3 on IMPLIRET: 17.9 to 16.4). Can you provide more insight into when and why synthesis degrades performance?

2. The threshold τ=0.3 is fixed across all experiments. How sensitive are results to this choice? Would adaptive, retriever-specific thresholds improve performance?

3. Could ARGUS be extended to iteratively refine embeddings during retriever training, rather than only at index time?

**Limitations:**

yes

**Strengths And Weaknesses:**

Soundness: The submission is technically sound. The proposed RPS metric is well-defined and grounded in a large-scale Wikidata-Wikipedia aligned dataset (7M entities). The authors demonstrate through LDA visualizations that low-RPS and high-RPS entities occupy distinguishable regions in embedding space, supporting their core hypothesis that blind spots are structural rather than random. The diagnostic probes achieve strong predictive performance (Pearson r ≈ 0.65-0.80), validating that retrievability risk is encoded in embedding geometry. Experimental validation is comprehensive, covering three benchmarks (BRIGHT, IMPLIRET, RAR-B) and eight neural retrievers with consistent improvements (averaging +3.4 nDCG@5 and +4.5 nDCG@10). However, the paper does not report computational costs for the ARGUS pipeline components (NER, probe inference, KB retrieval, LLM synthesis).


Presentation: The paper suffers from a few clarity and accessibility issues that limit its readability for those not deeply familiar with RAG systems. Some concepts are used without definition: "entities" are not defined in the introduction; "retrieval budgets," "context windows," and "top-k" appear without explanation in early sections; "mention span" is undefined in Section 3.2; and "NER" is not expanded in Section 5.1. The paper assumes readers understand how retrievers work in RAG pipelines. A brief technical primer defining query, entity, retrieval budget, embedding space, and semantic similarity would improve accessibility. Additionally, Figures 1 and 2 are too small to read effectively. In contrast, Figure 4 excellently illustrates the ARGUS pipeline. Evaluation metrics (nDCG@5/10) are presented in Table 2 without context for what constitutes good performance or typical score ranges, which again could be another topic that could use a quick primer. The entity embedding construction process, while detailed in Appendix B.4, could be explained more clearly in the main text.


Significance: The paper addresses an important problem in RAG systems: systematic retrieval blind spots that undermine trustworthiness. The shift from query-centric to entity-centric evaluation represents a meaningful contribution, as it enables pre-deployment auditing rather than only post-hoc diagnosis. ARGUS operates at index time without requiring retriever retraining or query rewriting, making it practically deployable. The consistent improvements across diverse retrievers and benchmarks suggest broad applicability. The two augmentation strategies (Document Expansion and LLM Synthesis) provide deployment flexibility for different index budget constraints. This work could influence how practitioners evaluate and improve retrieval systems, particularly in domains where entity coverage is critical.


Originality: The paper makes a novel contribution to the RAG space by introducing entity-level retrievability assessment through RPS, departing from existing query-centric, post-hoc evaluation methods. The insight that retrievability risk is predictable from embedding geometry enables proactive intervention rather than reactive fixes. The ARGUS pipeline (combining diagnosis via lightweight probes with targeted knowledge base augmentation) represents a creative solution to systematic blind spots. The work is well-positioned relative to prior literature on neural retrieval, RAG trustworthiness, and embedding geometry. The core conceptual advance is well-motivated through the initial RPS-LDA analysis demonstrating that blind spots correspond to structured regions in embedding space.

---

> ### Author Rebuttal · Authors · 2026-03-30
>
> We thank the reviewer for the positive assessment and insightful questions. We address the points below.
>
> ### (1) When does LLM synthesis degrade performance?
>
> We observe that synthesis can degrade performance when LLM-generated content introduces **hallucinated or imprecise context**, which acts as noise for the retriever.
>
> This effect is more pronounced in settings such as IMPLIRET, where documents tend to contain **more entities per context**, increasing the likelihood of incorrect or overly generalized augmentations, as shown below.
>
> **Table 1: Avg. extracted vs augmented entities per document (controls index growth; % = augmented/extracted).**
>
> |Datasets|BRIGHT|ImpliRet|RAR-b|
> |---|---|---|---|
> |**Extracted**||||
> ||1.01|3.12|0.43|
> |**Augmented**||||
> |bge-m3|0.96(95%)|2.41(77%)|0.41(96%)|
> |gritlm|0.83(82%)|2.48(80%)|0.36(83%)|
> |qwen3|0.86(85%)|1.51(49%)|0.37(87%)|
> |reasonir|0.58(57%)|1.61(52%)|0.25(58%)|
>
> In addition, we use a **shared, simple prompting template across retrievers and datasets**, as our goal was to study the effect of targeted augmentation rather than prompt engineering. As a result, the LLM is not always calibrated to dataset-specific structure or entity density, which can lead to variability across retrievers (e.g., JINA-V3).
>
> We expect that **more adaptive prompting strategies (e.g., conditioning on entity density or document structure)** could further reduce hallucination and improve robustness. In contrast, document expansion relies on retrieved KB evidence and is therefore more stable. We will clarify this trade-off in the revision.
>
> ### (2) Sensitivity to threshold τ
>
> We use a fixed threshold (tercile-based split) for simplicity and consistency across experiments. We agree that τ is a **tunable hyperparameter**, trading off coverage vs precision of augmentation.
>
> Our budget-based experiments (see Table 2 in the main paper) already provide a partial view of this sensitivity by showing performance across different augmentation levels. We expect that **retriever- or domain-specific calibration of τ** could further improve results, and we will clarify this in the revision.
>
> ### (3) Extension to retriever training
>
> We agree this is an interesting direction. While ARGUS is designed as a **pre-index, model-agnostic intervention**, its signals (e.g., low-RPS entities) could be used to guide **retriever training or fine-tuning**, for example by reweighting hard-to-retrieve entities or augmenting training data.
>
> We view this as a promising extension and will highlight it as future work.

---

> > ### Author Rebuttal · Reviewer_ULoR · 2026-04-04
> >
> > I thank the authors for their clarification of my concerns, which were mostly addressed. I would ask that the authors show an ablation study of the hyperparameter τ across different values to better understand this tradeoff between coverage and precision, which seems quite important to their final results.

---

> > > ### Author Response · Authors · 2026-04-04
> > >
> > > We thank the reviewer for this suggestion.
> > >
> > > **(τ Ablation and Coverage–Precision Tradeoff)**
> > >
> > > We thank the reviewer for this suggestion. We agree that the threshold τ controls the tradeoff between coverage (number of augmented entities) and precision (targeting true blind spots). We have conducted an ablation over τ and added these results to the revision.
> > >
> > > We evaluate τ ∈ {0.2,0.3,0.4,0.6,0.8,1.0}, where τ=1.0 corresponds to augmenting all entities and τ=0.0 is the baseline:
> > >
> > > | Model    | Threshold (τ)          | BRIGHT (3/10) | ImpliRet (2/2) | RAR-b (3/7) |
> > > |----------|-----------------------|---------------|----------------|-------------|
> > > | bge-m3   | Baseline (0.0)   | 10.43         | 21.05          | 19.13       |
> > > | 		   | 0.2                     | 12.52         | 30.01          | 19.81       |
> > > | 		   | ARGUS (0.3)     | 13.93         | 36.15          | 20.26       |
> > > | 		   | 0.4                     | 13.87         | 35.92          | 20.24       |
> > > | 		   | 0.6                     | 13.91         | 35.05          | 20.25       |
> > > | 		   | 0.8                     | 13.65         | 34.04          | 20.21       |
> > > | 		  | All (Entity) (1.0)| 13.31         | 33.45          | 20.19       |
> > > | ReasonIR | Baseline (0.0)   | 15.66         | 19.10          | 24.10       |
> > > | 		 | 0.2                     | 18.91         | 25.96          | 24.94       |
> > > | 		 | ARGUS (0.3)     | 21.13         | 30.60          | 25.53       |
> > > | 		 | 0.4                     | 20.98         | 30.43          | 25.51       |
> > > | 		 | 0.6                     | 20.59         | 30.21          | 25.42       |
> > > | 		 | 0.8                     | 20.23         | 29.42          | 25.37       |
> > > | 		 | All (Entity) (1.0) | 19.87         | 28.76          | 25.21       |
> > >
> > > *Table: Effect of τ on performance; lower τ increases precision, higher τ increases coverage but adds noise; τ≈0.3 performs best.*
> > >
> > > We observe:
> > > - Performance improves from τ=0.0→0.3.
> > > - Results are stable for τ∈[0.3,0.4].
> > > - Larger τ (≥0.6) degrades performance toward the all-entity regime.
> > >
> > > Overall, τ≈0.3 provides the best balance and performance is not highly sensitive in this range.
> > >
> > > We hope this resolves the reviewer’s concern.

---

### Official Review · Reviewer_Bu2P · 2026-03-12

**Soundness:** 4
**Presentation:** 3
**Significance:** 3
**Originality:** 4
**Overall Recommendation:** 5
**Confidence:** 5

**Summary:**

This paper investigates &quot;blind spots&quot; in neural retrievers used for Retrieval-Augmented Generation
(RAG) systems, instances where relevant entities are systematically failed to be retrieved due to
unfavorable embedding geometry rather than random chance. The authors argue that standard
query-centric evaluations fail to detect these intrinsic, entity-level risks prior to deployment.
To address this, the authors introduce the Retrieval Probability Score (RPS), a metric that
quantifies the likelihood of an entity being successfully retrieved under a specific budget (e.g., top-k)
when ranked against a large pool of disjoint &quot;neutral&quot; entities. Using a large-scale dataset derived
from Wikidata and Wikipedia, the study reveals that low retrievability is structurally encoded in the
embedding space: &quot;blind&quot; entities cluster in distinct, predictable geometric regions.
Building on these insights, the paper proposes ARGUS, a pipeline designed to diagnose and
remedy these blind spots before indexing. ARGUS uses lightweight probes to predict an entity&#39;s RPS
directly from its embedding. It then &quot;remedies&quot; high-risk entities (those with low predicted RPS) by
augmenting their documents with defining context retrieved from a reference knowledge base (e.g.,
Wikipedia), using methods such as document expansion or LLM-guided synthesis.

Contributions
The paper lists four primary contributions:
• Metric and Protocol (RPS): The authors introduce the Retrieval Probability Score (RPS)
and a domain-agnostic auditing protocol aligned with Wikidata and Wikipedia to assess
intrinsic entity-level retrievability risk.
• Geometric Predictability: They demonstrate that blind-spot risk is not random but is
encoded in the embedding representations, allowing for pre-index detection using lightweight
diagnostic probes without the need for expensive retrieval simulations.
• ARGUS Framework: They propose a practical &quot;diagnosis-to-remedy&quot; pipeline that
identifies high-risk entities via thresholding and mitigates their low retrievability by injecting
targeted external knowledge into the document index.

**Compliance With Llm Reviewing Policy:**

Affirmed.

**Key Questions For Authors:**

1. RPS and the diagnostic probes are developed and validated on a Wikidata/Wikipedia-
based corpus and a set of eight retrievers. How robust are these probes when deployed
on substantially different domains (e.g., biomedical, legal) or with very different
embedding models (e.g., sparse retrievers, multilingual encoders)?
---Evidence that the RPS-based diagnosis and probe models transfer across
domains/embedders, or a clear discussion of their limits, would strengthen the
significance and practicality of the work; if they are tightly coupled to the specific
Wikidata/Wikipedia setup, we would view the contribution as more specialized.
2. The experiments focus primarily on retrieval metrics (e.g., nDCG, Recall@k). Have you
evaluated whether ARGUS-style document expansion or LLM synthesis systematically
affects downstream generation quality, hallucination rates, or calibration in RAG systems
(e.g., via human evaluation or faithfulness metrics)?
--- Showing that remedies improve not just retrieval scores but end-task faithfulness
and robustness would further increase our assessment of significance; if
improvements are mainly metric-level without clear downstream gains, we would
see the impact as more incremental.

3. Document expansion and synthetic views enlarge the index and potentially shift the
embedding geometry. Could these interventions create new blind spots (e.g., entities
overshadowed by highly verbose expanded documents) or materially increase
latency/storage costs in realistic deployments?
---A more detailed analysis of these trade-offs (e.g., cost–benefit curves, sensitivity
to index size) would clarify deployment readiness; if significant new risks are
introduced, they should be more explicitly framed as limitations.

**Limitations:**

Yes. The authors discuss several limitations, including the dependence on a fixed reference
knowledge base (Wikipedia), the fact that remedies can increase index size and computational
cost, and that augmentation strategies may help or hurt depending on the retriever architecture.
It would strengthen the paper to more explicitly discuss (i) how far the RPS and probe models
can be trusted when moving beyond the Wikidata/Wikipedia setting (e.g., highly specialized
domains or multilingual corpora), (ii) whether targeted augmentation could inadvertently
introduce new blind spots or amplify biases present in the reference KB or LLM, and (iii) how
practitioners should balance storage/latency vs. robustness when deciding how aggressively to
apply ARGUS.

**Strengths And Weaknesses:**

Soundness
● Strengths: The submission is technically rigorous. The authors construct a large-scale,
domain-agnostic dataset using Wikidata and Wikipedia to define their Retrieval Probability
Score (RPS), ensuring the metric captures intrinsic geometric retrievability rather than query-
specific noise. The methodology for calculating RPS is robust, utilizing a &quot;neutral baseline&quot; of
disjoint entities to simulate controlled competition. The claim that blind spots are geometric
properties is well-supported by LDA visualizations showing clear clustering of low-RPS
entities. The use of diagnostic probes (XGBoost, Ridge, MLP) is validated through high
correlation (Pearson r ≈ 0.65–0.80) and low error rates, confirming predictability. The

proposed remedy, ARGUS, is empirically validated across three diverse benchmarks
(BRIGHT, IMPLIRET, RAR-B) and eight different neural retrievers, showing consistent
improvements.
● Weaknesses: The authors honestly acknowledge that their augmentation strategies
(document expansion vs. LLM synthesis) exhibit performance trade-offs depending on the
retriever architecture (e.g., JINA-V3 degrades with synthesis on certain tasks). Additionally,
the reliance on a single fixed Reference KB (Wikipedia) and a uniform prompt for all tasks
may be suboptimal for highly specialized domains.

Presentation
● Strengths: The paper is clearly written and well-structured, following a logical narrative arc:
Assess (RPS metric) $\rightarrow$ Detect (Diagnostic Probes) $\rightarrow$ Remedy
(ARGUS pipeline). Key concepts like &quot;blind spots&quot; and the distinction between random
misses and geometric failures are articulated clearly. Figure 2 effectively visualizes the core
insight of geometric predictability, and Figure 4 provides a lucid overview of the ARGUS
pipeline.
● Weaknesses: While the main narrative is clear, significant implementation details (e.g.,
specific probe hyperparameters, full per-task results) are relegated to the appendix, which is
common but requires the reader to consult supplementary material for full reproducibility.

Significance
● Strengths: The work addresses a critical and often overlooked failure mode in RAG
systems: the &quot;silent bottleneck&quot; where relevant information is structurally inaccessible to the
retriever. By shifting from post-hoc, query-centric evaluation to pre-index, entity-centric
auditing, the paper unlocks a new direction for ensuring RAG robustness before
deployment. The ARGUS framework offers practical utility, improving retrieval quality (e.g.,
+3.44 nDCG on BRIGHT) without the high cost of retraining the retriever.
● Weaknesses: The proposed remedies involve increasing the index size (either by
appending text or adding synthesized views), which creates a trade-off between
storage/compute efficiency and retrieval robustness that practitioners must manage.

Originality
● Strengths: The paper provides novel insights by formalizing the concept of &quot;retrieval blind
spots&quot; as predictable geometric regions rather than random errors. The introduction of RPS
as a metric for intrinsic retrievability is a new contribution. Furthermore, the method of
training lightweight probes to diagnose these blind spots pre-index, avoiding expensive
retrieval simulations, is an innovative application of existing techniques.
● Weaknesses: While the techniques for remedy (document expansion, synthetic data
generation) are established methods in IR, the novelty lies in their targeted application
based on uncertainty scoring to fix specific high-risk entities. The paper justifies this
combination well by demonstrating that targeted intervention is more effective than broad
application.

---

> ### Author Rebuttal · Authors · 2026-03-30
>
> We thank the reviewer for the very positive and thoughtful feedback. We are glad that the core contributions are found to be technically sound and practically meaningful. We address the questions below.
>
> ### (1) Generalization Across Domains and Embedding Models
>
> We agree that generalization beyond the Wikidata/Wikipedia setting is important. Our goal is to isolate a more fundamental phenomenon: **entity-level retrievability as a property of embedding geometry**, rather than a specific dataset.
>
> RPS is defined via relative ranking under embedding similarity; blind spots arise when relevant entities consistently fall outside top-k across query contexts. Thus, their *existence* is tied to embedding geometry, not a particular corpus.
>
> Empirically, we observe consistent behavior across **eight retrievers with diverse training objectives** and **multiple benchmarks (BRIGHT, ImpliRet, RAR-b)** spanning different reasoning settings. We view this diversity as practical evidence that the phenomenon is not dataset-specific.
>
> At the same time, **calibration** (e.g., probe thresholds, augmentation strategies, reference KB) may be domain-dependent. For specialized domains (e.g., biomedical, legal), domain-specific KBs and re-calibrated probes would likely improve performance. We will clarify this distinction in the revision.
>
> ### (2) Retrieval vs. Downstream RAG Utility
>
> We agree that downstream generation quality is important. Our work focuses on a *prior stage* of RAG: **ensuring relevant evidence is retrievable**.
>
> If a relevant document is not in top-k, it is **inaccessible to any downstream component**, regardless of generator or re-ranking strength. Thus, retrievability is a **necessary condition** for grounded generation.
>
> We study a pre-generation failure mode: systematic, geometry-induced blind spots. This complements prior work that improves performance *after retrieval*, which assumes relevant evidence is reachable.
>
> Evaluating downstream effects (e.g., faithfulness, hallucination) would be valuable. While beyond scope, we view this as a natural next step and will clarify this in the revision.
>
> ### (3) Cost, Trade-offs, and Potential Side Effects
>
> We agree that deployment trade-offs are important and analyze them explicitly.
>
> First, ARGUS introduces **minimal overhead**: augmentation is sparse (**<1 entity per document**), indicating targeted rather than uniform expansion (Table 1).
>
> **Table 1: Avg. extracted vs augmented entities per document (% = augmented/extracted).**
>
> |Datasets|BRIGHT|ImpliRet|RAR-b|
> |---|---|---|---|
> |**Extracted**||||
> ||1.01|3.12|0.43|
> |**Augmented**||||
> |bge-m3|0.96(95%)|2.41(77%)|0.41(96%)|
> |gritlm|0.83(82%)|2.48(80%)|0.36(83%)|
> |qwen3|0.86(85%)|1.51(49%)|0.37(87%)|
> |reasonir|0.58(57%)|1.61(52%)|0.25(58%)|
>
> Second, **budget-constrained augmentation** (prioritizing low-RPS entities) yields consistent gains even at 20–40%, with performance saturating as high-risk entities are covered (Table 2).
>
> **Table 2: Performance vs indexing budget (% vs baseline).**
>
> |Model|Budget(%)|BRIGHT(3/10)|ImpliRet(2/2)|RAR-b(3/7)|
> |---|---|---|---|---|
> |bge-m3|Baseline|10.43|21.05|19.13|
> ||20%|10.81(+3.6%)|21.82(+3.7%)|21.73(+13.6%)|
> ||40%|11.60(+11.2%)|23.63(+12.3%)|24.94(+30.4%)|
> ||60%|12.65(+21.3%)|26.10(+24.0%)|25.53(+33.4%)|
> ||80%|13.23(+26.8%)|26.64(+26.6%)|25.53|
> ||Full|13.93(+33.6%)|36.15(+71.8%)|25.53|
> |ReasonIR|Baseline|15.66|19.10|24.10|
> ||20%|16.57(+5.8%)|19.69(+3.1%)|25.13(+4.3%)|
> ||40%|18.92(+20.8%)|21.02(+10.1%)|25.53(+5.9%)|
> ||60%|21.13(+34.9%)|22.95(+20.2%)|25.53|
> ||80%|21.13|23.36(+22.3%)|25.53|
> ||Full|21.13|30.60(+60.2%)|25.53|
>
>
> Third, in a **no index growth setting** (LLM replacement), ARGUS still yields consistent improvements (Table 3).
>
>
> **Table 3: No index growth (replacement vs synthesis).**
>
> |Model|Method|BRIGHT(3/10)|ImpliRet(2/2)|RAR-b(3/7)|
> |---|---|---|---|---|
> |bge-m3|Baseline|10.43|21.05|19.13|
> ||ARGUS(LLM Replacement)|13.48|23.72|20.06|
> ||ARGUS(LLM Synthesis)|13.90|24.50|20.13|
> |ReasonIR|Baseline|15.66|19.10|24.10|
> ||ARGUS(LLM Replacement)|17.86|21.15|25.49|
> ||ARGUS(LLM Synthesis)|18.26|21.70|25.53|
>
> Overall, ARGUS applies **targeted, entity-aware augmentation** on high-risk entities, avoiding noise from uniform expansion while maintaining efficiency.
>
> ### (4) Limitations and Practical Considerations
>
> We will expand limitations as suggested. Key considerations include:
> - **Domain dependence**: need for domain-specific KBs and calibration
> - **Retriever sensitivity**: varying responses to expansion vs synthesis
> - **Cost–robustness trade-offs**: controllable via augmentation budgets
>
> We thank the reviewer again and will incorporate these clarifications to further strengthen the paper!

---

> > ### Author Rebuttal · Reviewer_Bu2P · 2026-04-05
> >
> > No more Concern

---

### Official Review · Reviewer_m28P · 2026-03-13

**Soundness:** 2
**Presentation:** 2
**Significance:** 2
**Originality:** 2
**Overall Recommendation:** 2
**Confidence:** 5

**Summary:**

The authors argue that existing dense retrievers that rely on embedding-based similarity to retrieve relevant documents for a query have blindspots where they miss entity matches and by consequence, the corresponding relevant documents due to low semantic similarity to query. The authors propose a measure - retrieval probability score (RPS) to detect such blind-spots from embedding geometry alone. Then, detecting cases of low-RPS, authors propose the pipeline ARGUS which employs knowledge augmentation from a knowledge base constructed from Wikipedia to retrieve such documents which mitigates the blind-spot issue.

**Compliance With Llm Reviewing Policy:**

Affirmed.

**Key Questions For Authors:**

Please refer weaknesses section.
Additional queries:
Do you use only 1-hop entity relations. Were additional hop traversals explored? Can you elaborate on the rationale behind this design choice. Won't additional hops be useful for reasoning-intensive retrieval tasks like BRIGHT ?

Can you also please add BM25 + RM3 as a baseline. BM25 is usually a very strong baseline even for reasoning-intensive retrieval tasks like BRIGHT and is more powerful with RM3 based expansion.

**Limitations:**

yes

**Strengths And Weaknesses:**

Strengths:

1. The work tackles an interesting problem with regards to first-stage retrieval issues, which is also a long-standing problem investigated in the information retrieval field.
2. The authors experiment on diverse datasets including recent reasoning-intensive retrieval tasks like BRIGHT.
3. The proposed RPS approach is intuitive and the geometric analysis further helps capture the intuition behind how RPS could be used to diagnose such blindspots.

Weaknesses:

Knowledge graph-based ranking beyond semantic matching has been explored previously [1,5]. I believe this is an important baseline and authors must perform a comprehensive literature survey. I believe the current comparisons are weak and does not include relevant baselines from different sub-fields in IR as detailed in my points below. Though experiments are performed on diverse datasets, I believe the current work ignores relevant literature in IR that aims to solve such retrieval blindspots.
Authors primarily motivate their approach on two fronts - one to capture documents that may be relevant but have low similarity due to retriever blindspots, resulting in low semantic similarity to the query. Hence, they posit that detecting these blindspots and mitigating the issue using ARGUS would also result in practical recall improvements at lower depths, which is practical for RAG applications. However, authors do not perform a comprehensive survey of similar works in Information retrieval which aim to solve this issue. Adaptive retrieval approaches [2,3,4] aim to solve such blindspots which they call as recall boundedness, where documents that are relevant may be missed due to blindspots in lexical or dense retrieval and are never resurfaced during ranking stage. Hence, they focus on forming an offline corpus graph that stores the k-nearest relevant documents for each document in the corpus. This graph is created offline and is a one-time cost. During inference, the adaptive retrieval/ranking approach traverses this corpus graph and uses cross-encoder signals to prioritize relevant documents that are usually missed by the retriever. This approach also does not add much to latency during inference and provides significant recall gains, especially at lower depths. Their efficacy for RAG has also been documented [4]. The adaptive retrieval approaches are primarily based on clustering hypothesis, which states that closely associated or similar documents tend to be relevant to the same information requests. Hence, this document-document affinity and other clever signals learned in such approaches help solve retriever blindspots. I believe this is a very relevant baseline.
I believe query reformulation approaches like GenQR, genQRensemble, query2doc are also relevant here as they help solve the vocabulary mismatch gap and also help generate reformulations of the original query, especially relevance feedback-based approaches [6], that help bridge the blindspots of the retriever and helps retrieve relevant documents. I believe authors should discuss this in related work and compare to or argue the case for not employing such approaches which are far simpler yet shown to be very effective.
Though the paper tackles an important problem, I believe the current version of the paper ignores relevant prior work that has been well researched upon in IR field as detailed in my above points.



[1] Jinyuan Fang, Zaiqiao Meng, and Craig Macdonald. 2023. KGPR: Knowledge Graph Enhanced Passage Ranking. In Proceedings of the 32nd ACM International Conference on Information and Knowledge Management (CIKM '23). Association for Computing Machinery, New York, NY, USA, 3880–3885. https://doi.org/10.1145/3583780.3615252
[2] Mandeep Rathee, Sean MacAvaney, and Avishek Anand. 2025. Quam: Adaptive Retrieval through Query Affinity Modelling. In Proceedings of the Eighteenth ACM International Conference on Web Search and Data Mining (WSDM '25). Association for Computing Machinery, New York, NY, USA, 954–962. https://doi.org/10.1145/3701551.3703584
[3] Sean MacAvaney, Nicola Tonellotto, and Craig Macdonald. 2022. Adaptive Re-Ranking with a Corpus Graph. In Proceedings of the 31st ACM International Conference on Information & Knowledge Management (CIKM '22). Association for Computing Machinery, New York, NY, USA, 1491–1500. https://doi.org/10.1145/3511808.3557231
[4] Adaptive Retrieval for Reasoning-Intensive Retrieval
Jongho Kim, Jaeyoung Kim, Seung-won Hwang, Jihyuk Kim, Yu Jin Kim, Moontae Lee
[5] Ikuya Yamada, Akari Asai, Hiroyuki Shindo, Hideaki Takeda, and Yuji Matsumoto.2020. LUKE: Deep Contextualized Entity Representations with Entity-awareSelf-attention. In Proceedings of the 2020 Conference on Empirical Methods inNatural Language Processing. 6442–6454
[6] Iain Mackie, Shubham Chatterjee, and Jeffrey Dalton. 2023. Generative Relevance Feedback with Large Language Models. In Proceedings of the 46th International ACM SIGIR Conference on Research and Development in Information Retrieval (SIGIR '23). Association for Computing Machinery, New York, NY, USA, 2026–2031. https://doi.org/10.1145/3539618.3591992

---

> ### Author Rebuttal · Authors · 2026-03-30
>
> We thank the reviewer for the thoughtful feedback and for highlighting relevant work. We appreciate these pointers and address the concerns below.
>
> ### 1. Positioning vs. Prior Work, Problem Definition, and Evaluation Completeness
>
> Thanks for emphasizing relevant IR literature, including knowledge-graph-based retrieval, adaptive retrieval, and query reformulation. We agree these methods improve recall, but operate at a different stage.
>
> Prior work focuses on **retrieval-time improvements**, via query-side (e.g., reformulation, feedback) or document-side augmentation, assuming a fixed retriever and improving its outputs [1,2]. As such, they are not direct baselines for our setting, which studies failures *prior* to retrieval.
>
> In contrast, we study whether the **retriever exhibits intrinsic, query-agnostic blind spots** in its embedding space. Existing approaches assume relevant evidence is reachable; we explicitly model when it is not, i.e., when relevant documents fail to appear within top-k.
>
> To our knowledge, prior work does not model retriever blind spots in a query- and dataset-agnostic way, nor using external reference corpora independent of the target dataset [1,2].
>
> Concretely, we model **entity-level retrievability as a property of the retriever under a given top-k budget (capacity)**, independent of any query or dataset. Our Retrieval Probability Score (RPS) enables **pre-index diagnosis of retriever-specific failure modes from embedding geometry**, without query-time intervention.
>
> Unlike adaptive retrieval methods, which expand from initially retrieved documents, our approach is **query- and document-agnostic**, modeling failures *before* observing any input. If a relevant document is not retrieved within top-k, downstream methods cannot access it. Rather than recovering missed evidence, we **diagnose failures and apply targeted, non-uniform corrections** only where needed. Our approach is complementary and can be combined with retrieval-time methods.
>
> This is reflected in Table 1: on average, <1 entity per document is augmented, indicating **precise correction**, not uniform expansion.
>
> **Table 1: Avg. extracted vs augmented entities per document (controls index growth; % = augmented/extracted).**
> |Datasets|BRIGHT|ImpliRet|RAR-b|
> |---|---|---|---|
> |**Extracted**||||
> ||1.01|3.12|0.43|
> |**Augmented**||||
> |bge-m3|0.96(95%)|2.41(77%)|0.41(96%)|
> |gritlm|0.83(82%)|2.48(80%)|0.36(83%)|
> |qwen3|0.86(85%)|1.51(49%)|0.37(87%)|
> |reasonir|0.58(57%)|1.61(52%)|0.25(58%)|
>
> ARGUS is a **diagnosis-to-remedy pipeline**: the core contribution is retriever-centric auditing, with augmentation as a downstream step.
>
> We evaluate across datasets and settings, consistently observing improvements without modifying the retriever or query.
>
> We will clarify this positioning in the paper.
>
> ### 2. BM25 Baseline
>
> Thanks for this suggestion. We agree that BM25 (and BM25+RM3) are strong lexical baselines.
>
> However, our work targets **dense retrievers and embedding geometry**, as both RPS and ARGUS rely on learned representations. RPS is defined over embedding space and captures **retriever-specific failure modes**, not defined for lexical methods.
>
> Our goal is not to compare paradigms, but to isolate **retriever-internal failure modes within embedding models**, not addressed by retrieval-time methods.
>
> We will clarify this scope.
>
> ### 3. 1-hop vs. Multi-hop
>
> We thank the reviewer for this question. Our focus is on **entity-level retrievability**, i.e., whether a retriever retrieves a relevant document.
>
> Multi-hop retrieval aggregates across documents [1]. However, if relevant documents are not retrieved due to blind spots, multi-hop reasoning cannot be applied. Additional hops may help, but do not resolve cases where relevant entities are never retrieved.
>
> We focus on the 1-hop setting to isolate **intrinsic retriever limitations**, and will clarify this.
>
> ### 4. Dataset Choice (BRIGHT, ImpliRet, RAR-b)
>
> We thank the reviewer for this question. These datasets are designed to **reduce lexical overlap**, emphasizing semantic retrieval.
>
> In high-overlap settings, lexical signals can dominate and mask limitations of dense retrievers, whereas low-overlap settings require semantic matching [1,3]. Thus, these datasets better expose **retriever failures due to embedding mismatch**, aligning with our focus.
>
> These datasets provide a suitable testbed for analyzing **retriever limitations beyond lexical matching**.
>
> We thank the reviewer again for the valuable feedback. We will incorporate these clarifications to improve positioning, related work, and evaluation clarity.
>
> [1] Gao et al., *Retrieval-Augmented Generation for Large Language Models: A Survey*, 2023.
>
> [2] Lin et al., *A Survey of Retrieval-Augmented Generation for LLMs*, 2024.
>
> [3] Karpukhin et al., *Dense Passage Retrieval for Open-Domain Question Answering*, 2020.

---

### Official Review · Reviewer_2o9S · 2026-03-13

**Soundness:** 3
**Presentation:** 3
**Significance:** 2
**Originality:** 3
**Overall Recommendation:** 4
**Confidence:** 4

**Summary:**

This paper studies entity-level retrieval blind spots in neural retrievers used in RAG. The authors define blind spots as entities that are semantically relevant to a query but consistently missed by the retriever due to unfavorable embedding geometry. It introduces the Retrieval Probability Score, a metric computed from a Wikidata-Wikipedia alignment protocol that estimates how often an entity is retrieved within top-k under controlled neutral competition, and trains lightweight probes to predict RPS from entity embeddings before indexing. Based on these predictions, the paper proposes ARGUS, an index-time augmentation pipeline that identifies high-risk entities and augments documents with KB-derived context via either document expansion or LLM synthesis. Experiments report average improvements over the original indexing setup.

**Compliance With Llm Reviewing Policy:**

Affirmed.

**Final Justification:**

Author rebuttal addressed major concerns. If the author can incorporate the discussion and new results, I would give a score above borderline.

**Key Questions For Authors:**

see weakness

**Limitations:**

yes

**Strengths And Weaknesses:**

Strengths:

- The perspective to improve retrieval is interesting: it shifts from query-centric, post-hoc evaluation to an entity-centric, pre-index auditing paradigm. Identifying what a retriever will systematically miss before runtime offers a scalable way to improve RAG trustworthiness and also accuracy.

- Experiments are comprehensive. The empirical scope is broader than many papers in this area. Evaluating eight retrievers on three recent retrieval benchmarks is a meaningful effort and makes this paper easy to follow.

- The interpretability and visualization experiments are appreciated, which offer new insights and views towards this problem. However, the geometric evidence is weaker than advertised: Figure 2 uses LDA with labels defined by RPS terciles. LDA is specifically designed to separate labeled classes, so the visual separability is not surprising.

Weaknesses:

- Method efficiency. The proposed method includes extended steps compared to the general retriever, including NER, RPS, augmentation. It is unclear whether the performance improvement is worth the effort. Add such experiments would improve the validity.

- Ablation studies should align more with each contribution. The paper argues that gains come from detecting and remedying blind spots, but Table 2 only compares Original vs ARGUS. There is no comparison to untargeted KB expansion, random-entity augmentation, or augmenting all named entities.

- The paper remains too retrieval-centric given the broader robust-RAG framing in writing. The current results still do not tell us which remedy provides more downstream-useful evidence rather than simply improving matchability. A small answer-side utility analysis (e.g., a SePer / Infogain-RAG metric), or at least add a discussion in the utility perspective with recent strong search practices like Self-RAG / search-R1, would substantially strengthen this paper in the ful RAG cycle.

---

> ### Author Rebuttal · Authors · 2026-03-30
>
> Thanks for the feedback and positive assessment.
> ### (1) LDA (Fig. 2)
> LDA is supervised (labels are RPS terciles: low/mid/high). Fig. 2 is a sanity check, not a claim. Results rely on Table1 (RPS predictability).
> ### (2) Efficiency
> Thanks; ARGUS runs entirely pre-index (offline) and does not modify inference-time retrieval (same retriever/scoring). The only change is a potential increase in indexed views.
> Table 1 shows that this overhead is sparse: <1 augmented entity per document (e.g., 0.25-0.58 for ReasonIR), implying limited index growth.
>
> **Table 1: Avg. extracted vs augmented entities per document (controls index growth; % = augmented/extracted).**
> |Datasets|BRIGHT|ImpliRet|RAR-b|
> |---|---|---|---|
> |**Extracted**||||
> ||1.01|3.12|0.43|
> |**Augmented**||||
> |bge-m3|0.96(95%)|2.41(77%)|0.41(96%)|
> |gritlm|0.83(82%)|2.48(80%)|0.36(83%)|
> |qwen3|0.86(85%)|1.51(49%)|0.37(87%)|
> |reasonir|0.58(57%)|1.61(52%)|0.25(58%)|
>
> We evaluate **budget-constrained augmentation** (Table 2) by selecting the lowest-RPS entities. ARGUS yields consistent gains even under tight budgets(20-40%), with saturation once most high-risk entities are covered.
>
> **Table 2: Performance vs indexing budget (lowest-RPS prioritized; % vs baseline).**
> |Model|Budget(%)|BRIGHT(3/10)|ImpliRet(2/2)|RAR-b(3/7)|
> |---|---|---|---|---|
> |bge-m3|Baseline|10.43|21.05|19.13|
> ||20%|10.81(+3.6%)|21.82(+3.7%)|21.73(+13.6%)|
> ||40%|11.60(+11.2%)|23.63(+12.3%)|24.94(+30.4%)|
> ||60%|12.65(+21.3%)|26.10(+24.0%)|25.53(+33.4%)|
> ||80%|13.23(+26.8%)|26.64(+26.6%)|25.53|
> ||Full|13.93(+33.6%)|36.15(+71.8%)|25.53|
> |ReasonIR|Baseline|15.66|19.10|24.10|
> ||20%|16.57(+5.8%)|19.69(+3.1%)|25.13(+4.3%)|
> ||40%|18.92(+20.8%)|21.02(+10.1%)|25.53(+5.9%)|
> ||60%|21.13(+34.9%)|22.95(+20.2%)|25.53|
> ||80%|21.13|23.36(+22.3%)|25.53|
> ||Full|21.13|30.60(+60.2%)|25.53|
>
> We also consider **no index growth** (Table 3) by replacing each document with its LLM-augmented version. Even without increasing indexed documents, ARGUS improves performance.
>
> **Table 3: LLM augmentation without increasing index size (replacement vs synthesis).**
> |Model|Method|BRIGHT(3/10)|ImpliRet(2/2)|RAR-b(3/7)|
> |---|---|---|---|---|
> |bge-m3|Baseline|10.43|21.05|19.13|
> ||ARGUS(LLM Replacement)|13.48|23.72|20.06|
> ||ARGUS(LLM Synthesis)|13.90|24.50|20.13|
> |ReasonIR|Baseline|15.66|19.10|24.10|
> ||ARGUS(LLM Replacement)|17.86|21.15|25.49|
> ||ARGUS(LLM Synthesis)|18.26|21.70|25.53|
>
> Across all settings ARGUS consistently improves retrieval quality, yielding strong gains at low or zero added cost.
> ### (3) Ablations
> Thanks, comparisons to un-targeted augmentation are essential.
> We evaluate strategies along two axes: entity grounding (entity-aware vs un-targeted) and selection (random/all/targeted). Results in Table 4.
> We observe (for each dataset, M entities are extracted via NER, and N ⊆ M are eligible for augmentation from the KB):
> - **Un-targeted (N random KB docs, independent of document entities):** little/no gain, may degrade
> - **Random (N random entities, without RPS-based selection):** moderate gains, but suboptimal
> - **All entities (all M entities):** improves, but below ARGUS (adds noise)
> - **ARGUS (subset of N with low RPS):** best; gains depend on **correct entity selection**
> Thus, both *entity grounding* and *targeted selection* are necessary.
>
> **Table 4: Effect of entity grounding and selection strategy on retrieval.**
> |Model|Method|Entity-Aware|Selection|BRIGHT(3/10)|ImpliRet(2/2)|RAR-b(3/7)|
> |---|---|---|---|---|---|---|
> |bge-m3|Baseline|–|–|10.43|21.05|19.13|
> ||Un-targeted(Random)|✗|None|11.21|21.12|19.52|
> ||Random(Entity)|✓|Random|11.43|31.64|20.12|
> ||All(Entity)|✓|All|13.31|33.45|20.19|
> ||ARGUS|✓|Low-RPS|13.93|36.15|20.26|
> |ReasonIR|Baseline|–|–|15.66|19.10|24.10|
> ||Un-targeted(Random)|✗|None|15.81|19.04|24.17|
> ||Random(Entity)|✓|Random|18.03|25.54|24.78|
> ||All(Entity)|✓|All|19.87|28.76|25.21|
> ||ARGUS|✓|Low-RPS|21.13|30.60|25.53|
>
> Un-targeted: unrelated; entity-aware: entity-level; ARGUS: targeted(low-RPS).
> ### (4) Retrieval vs RAG
> Thanks, we agree that downstream RAG utility is important. Our focus is complementary: ensuring relevant evidence is retrievable.
> Prior work distinguishes query-side (inference-time; outside our scope) and document-side methods [1,2], where the latter typically rely on uniform expansion (e.g., document/query augmentation).
> In contrast, our method is both query-agnostic and document-agnostic: we model retriever-specific failures via a top-k-aware metric (RPS) to identify systematic blind spots, and address them with targeted (non-uniform) augmentation.
> This retriever-centric, failure-driven approach, independent of specific queries or datasets, has not been explored to our knowledge.
> We will clarify this scope.
>
> [1] Gao et al., Retrieval-Augmented Generation for Large Language Models: A Survey, 2023.
>
> [2] Lin et al., A Survey of Retrieval-Augmented Generation for LLMs (Document vs Query-side methods), 2024.
>
> We hope this clarifies the concerns; we are happy to provide further details.

---

> > ### Author Rebuttal · Reviewer_2o9S · 2026-04-04
> >
> > Thanks for the reply. Most of my concerns are addressed.

---

### Decision · Program_Chairs · 2026-04-30

**Decision:**

Accept (regular)

**Comment:**

- This paper identifies an important failure mode in neural retrieval for RAG systems—entity‑level retrieval blind spots—and proposes a principled, pre‑index diagnosis and remediation framework (RPS + ARGUS). Reviewers generally found the approach technically sound, and post‑rebuttal, most indicated that concerns related to efficiency, augmentation tradeoffs, and threshold sensitivity were largely addressed.
- One remaining concern is the lack of comparison to strong lexical baselines such as BM25. While such comparisons would strengthen practical positioning, their absence limits comparative evaluation rather than undermining the validity of the core contribution and appears feasible to address in revision.
- Overall, I recommend Weak Accept, noting that inclusion of standard baselines could warrant a higher assessment.